# General technoeconomic analysis for electrochemical coproduction coupling carbon dioxide reduction with organic oxidation

Jonggeol Na [1,7], Bora Seo[1], Jeongnam Kim[1,2], Chan Woo Lee [3], Hyunjoo Lee[1,4], Yun Jeong Hwang [1,4,5], Byoung Koun Min [1,6], Dong Ki Lee [1], Hyung-Suk Oh [1,4]* & Ung Lee [1,4,6]*

Electrochemical processes coupling carbon dioxide reduction reactions with organic oxidation reactions are promising techniques for producing clean chemicals and utilizing renewable energy. However, assessments of the economics of the coupling technology remain questionable due to diverse product combinations and significant process design variability. Here, we report a technoeconomic analysis of electrochemical carbon dioxide reduction reaction–organic oxidation reaction coproduction via conceptual process design and thereby propose potential economic combinations. We first develop a fully automated process synthesis framework to guide process simulations, which are then employed to predict the levelized costs of chemicals. We then identify the global sensitivity of current density, Faraday efficiency, and overpotential across 295 electrochemical coproduction processes to both understand and predict the levelized costs of chemicals at various technology levels. The analysis highlights the promise that coupling the carbon dioxide reduction reaction with the value-added organic oxidation reaction can secure significant economic feasibility.

[1] Clean Energy Research Center, Korea Institute of Science and Technology (KIST), 02792 Seoul, Republic of Korea. [2] School of Chemical and Biological Engineering, Seoul National University, Gwanak-ro 1, Gwanak-gu, 08826 Seoul, Republic of Korea. [3] Department of Chemistry, Kookmin University, 02707 Seoul, Republic of Korea. [4] Division of Energy and Environmental Technology, KIST School, Korea University of Science and Technology (UST), 02792 Seoul, Republic of Korea. [5] Department of Chemical and Biomolecular Engineering, Yonsei University, 03722 Seoul, Republic of Korea. [6] Green School, Korea University, 145 Anam-ro, Seongbuk-gu, 02841 Seoul, Republic of Korea. [7] Present address: Department of Chemical Engineering, Carnegie Mellon University, 5000 Forbes Ave, Pittsburgh, PA 15213, USA. *email: hyung-suk.oh@kist.re.kr; ulee@kist.re.kr

The electrochemical conversion of $CO_2$ to produce valuable chemicals is an excellent potential technology that satisfies carbon emission reduction and stores electricity obtained from renewable energy sources in chemical form. Point sources with intensive $CO_2$ concentrations are located at power plants, cement production, and petrochemical facilities where carbon capture and utilization can be accomplished to create carbon-neutral cycles[1,2]. The products of the $CO_2$ reduction reaction ($CO_2$RR), such as CO, syngas, methanol, and ethylene, can be used as reagents to create many chemicals, plastics, and transportation fuels[3], therefore providing a possibility to replace fossil fuel-based processes. In addition, formic acid has become attractive as a safe and ecofriendly liquid hydrogen carrier for the hydrogen economy[4]. $CO_2$RR technology has advanced rapidly in recent years, ranging from catalyst development to the engineering of electrolytes and cell systems. At the lab scale, the production of ethylene from $CO_2$ can be achieved with 70% Faraday efficiency (FE) at a high current density of 100 mA cm$^{-2}$ [5]. Given that the $CO_2$RR can proceed via complex reaction pathways[6], such outstanding performance indicates the technological possibility of practical implementation.

The economic feasibility of $CO_2$RR technology has been evaluated via several technoeconomic analyses (TEAs) in the last 10 years. Important insights into the favorable products and required performances from an economic point of view have been reported[7–9], while critical opinions have been voiced owing to the considerable production cost relative to market price despite simplified modeling[10,11]. Previous works have focused on only the cathode reaction under the assumption that water oxidation to produce oxygen gas occurs at the anode, which means that the anode reaction creates minimal value and is included in only the operational cost. Recently, new strategies have been proposed involving electrolysis in combination with other organic oxidation reactions (OORs) to produce value-added products rather than oxygen gas. For example, the oxidation of biomass-derived 5-hydroxymethylfurfural (HMF) to 2,5-furandicarboxylic acid (FDCA), a building block for various plastics, can be incorporated as an anode reaction with a very low overpotential[12]. Verma et al[13]. have proved that the anodic electro-oxidation of glycerol substituting $O_2$ evolution reaction (OER), which leads to lower electricity consumption by up to 53%. The economic feasibility of $CO_2$RR technology could be assessed more profitably and reasonably if both the cathode and anode reactions were considered.

Here, we conducted an extensive TEA of 295 electrochemical coproduction combinations (i.e., 16 cathode and 18 anode reactions with 7 cascade processes) to evaluate the economic feasibility of $CO_2$RR technology combined with value-added OOR and find a profitable candidate combination. The profitability was estimated based on the relative ratio of levelized costs of chemicals (LCC) to market price, where the LCC represents the minimum selling price without a margin. The analysis results reveal that the profitability index significantly depends on the type of OOR rather than the type of $CO_2$RR. Furthermore, coupling $CO_2$RR with biomass oxidation in an electrochemical system can promise substantial revenue. An automatic process synthesis framework was developed to analyze a large number of coproduction processes in which all the factors that can affect the production cost, including electrolyzer systems, separation processes, recycling systems, and various utility systems, were thoroughly considered, thereby ensuring analytical reliability. Furthermore, we used global sensitivity analysis to deconvolute the contributions of current density, FE, and overpotential to the LCC, which indicated which parameter must be preferentially addressed to achieve profitability.

## Results

**Electrochemical coproduction.** To run an electrolysis cell, two half-reactions, oxidation and reduction, should be paired to create a complete reaction. Herein, we define the electrochemical coproduction as a paired electrolysis that both cathodic $CO_2$RR and anodic OOR produce chemicals with reasonable market values. There are four types of electrochemical coproduction: parallel, convergent, divergent, and linear paired electrolysis (Fig. 1). Parallel paired electrolysis features the simultaneous occurrence of two unrelated half-reactions in a divided cell. The most well-known, industrially established example is the chlor-alkali process, wherein chlorine and sodium hydroxide are produced at the anode and cathode, respectively[14]. Interestingly, small-scale $CO_2$RR–OOR process demonstration regarding the oxidative condensation through molecular electrocatalysts belongs to a parallel type[15]. The parallel paired electrolysis can be very challenging if significant differences exist between half reaction operating conditions (i.e., solvent, pH, temperature, etc.) and the different operating conditions may cause expensive electrolyzer design and fabrication cost. We summarize electrolysis conditions of both cathodic and anodic products in Supplementary Table 1. Convergent paired electrolysis is designed to produce a single product from the reaction between intermediates formed in the cathode and anode in an undivided cell. An example is the electrosynthesis of cyanoacetic acid, which uses $CO_2$ and acetonitrile as substrates (Fig. 1a, top right)[16]. Divergent paired electrolysis is performed with a common starting substrate in both electrodes, leading to different products. For example, the electrolysis of dienes in the cathode and anode simultaneously produces diacids and diol derivatives, respectively (Fig. 1a, bottom left)[17]. When the same product is produced from the same substrate via different electrochemical reactions, the process is classified as linear paired electrolysis. The electrochemical conversion of dibutyl N-hydroxylamine to N-butylidenbutylamine N-oxide is a linear paired electrolysis process (Fig. 1a, bottom right)[18].

The pairing of two efficient half-reactions is also essential in terms of energy savings. As shown in Fig. 1b, the overall electrical energy required to perform electrolysis is determined by the sum of the cathodic and anodic energies required to conduct each half-reaction. Thus, additional energy must inevitably be consumed in the counter reaction to perform a desirable half-reaction. Therefore, effective pairing not only improves the atom economy but also increases energy efficiency (Figs. 1b and 2a, b: EG, methanol, n-propanol, and Cu product family). In addition, such electrochemical synthesis of organics is more environmentally friendly than conventional nonelectrochemical industrial processes. The electrochemical technique is usually performed at ambient pressure and temperature and minimizes the use of hazardous chemical oxidants. Importantly, since several $CO_2$RR and OOR can practically guarantee a high FE even at current technology levels (Fig. 2a, c), $CO_2$RR–OOR technology can have a very promising outlook.

Here, we selected 16 cathodic reactions as a candidate including $CO_2$RR and $H_2$ evolution reaction (HER) and the target products were divided into three categories based on FE, partial current density, and cell potential characteristics (Table 1). We also considered 18 anodic reactions including OOR and OER then divided these candidates into three categories based on the source of raw material (Table 1). Among the several electrolyzer and paired electrolysis concepts, we considered a commercial electrolyzer with parallel paired electrolysis (Fig. 1a and Supplementary Fig. 1b). Although more advanced device configurations (e.g., Supplementary Fig. 1) are being actively studied, they still remain in lab-scale development. The detailed information with appropriateness of assumption including half-cell reaction, standard reduction potential, and overpotential of

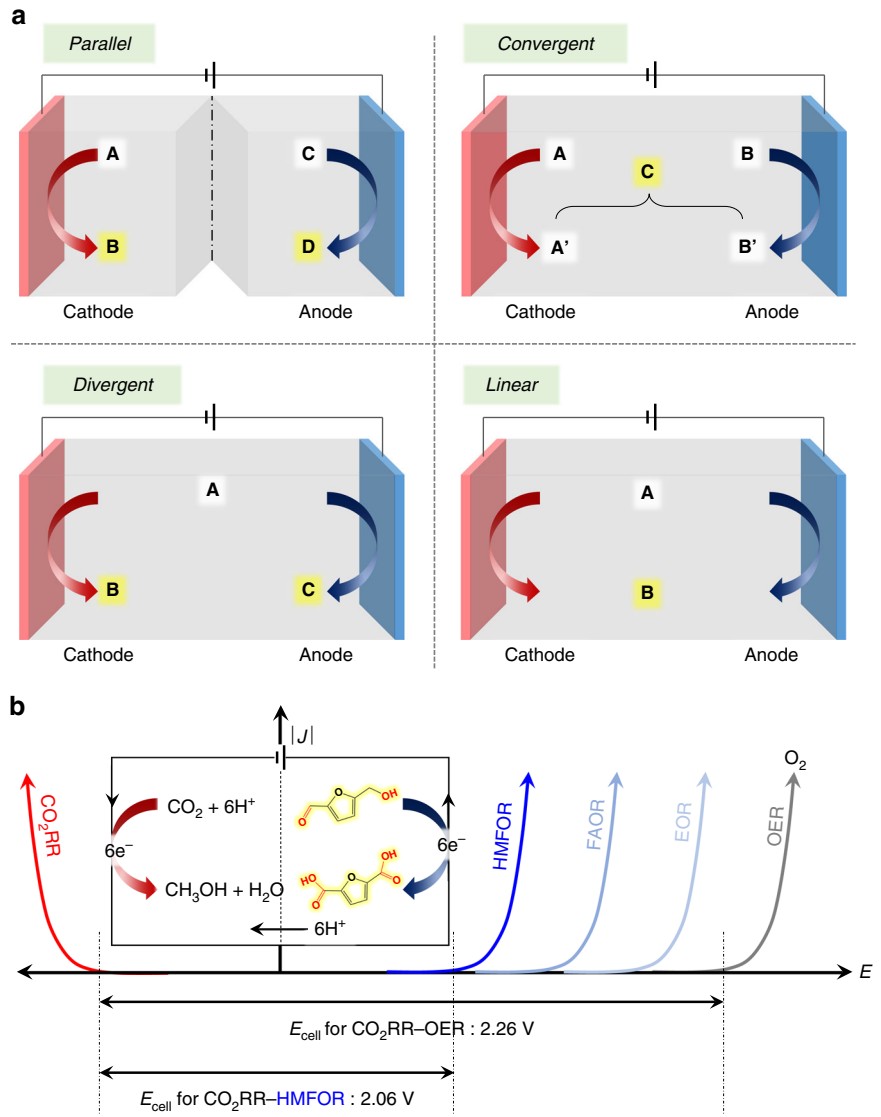

**Fig. 1** Schematic illustration of the electrochemical coproduction system. **a** Parallel, convergent, divergent, and linear paired electrolysis. **b** I–V curves and required potentials at the cathode and anode for electrolysis

candidates, market analysis, and electrolyzer devices are given in the Supplementary Methods.

**Process systems**. To analyze a large number of CO₂RR–OOR process combinations, we developed an automatic process synthesis framework comprising process flowsheet generation, calculation, and TEA (Fig. 3a and Supplementary Fig. 2). Figure 3b illustrates the superstructure considered in this study. The superstructure includes every possible process design of CO₂RR–OOR coproduction, product separation, and recycling options. The superstructure is reduced to the appropriate process design (Supplementary Fig. 3a–h) according to the produced chemicals using algorithm in Table 2, and the process structure is then transferred to a process simulator for automatic process flowsheet generation. Note that structure generation and flowsheet evaluation are performed automatically within the platform; thus, a relatively large number of CO₂RR–OOR processes can be efficiently evaluated. To illustrate the automatic platform, we provide an electrochemical coproduction process for the production of CO/2-furoic acid in the Supplementary Notes.

A detailed schematic of the CO₂RR–OOR superstructure is shown in Fig. 3b. The proposed superstructure generally consists

of the cathode and anode production/separation parts. We also considered a cascade production process in which the cathode products are supplied as anode reactants to create the final products (e.g. methanol and ethylene glycol). In the cathode production part, the CO₂-saturated electrolyte in the mixing tank is pumped to the cathode of the electrochemical reactor and receives electrons to produce various target products. Each target material requires an appropriate separation process because of its different physical and chemical properties. We adopted the separation method that is currently being used for commercial purposes. The gas/liquid mixture from the reactor is separated by using a flash separator. After that, gas/gas separation and liquid/ liquid separation are employed according to necessity. The cascade splitter is activated to use the products from the cathode as organic raw materials for the anode. For example, methanol generated at the cathode may be used as a raw material for anode oxidation to produce formic acid. In the anode, various organic compounds are used as raw materials for valorization through organic oxidation in the coproduction scheme. These organic compounds are produced through either biomass processes or other separate preprocesses. Electrolytes and organic compounds are mixed and then moved to the anode side. The solution is

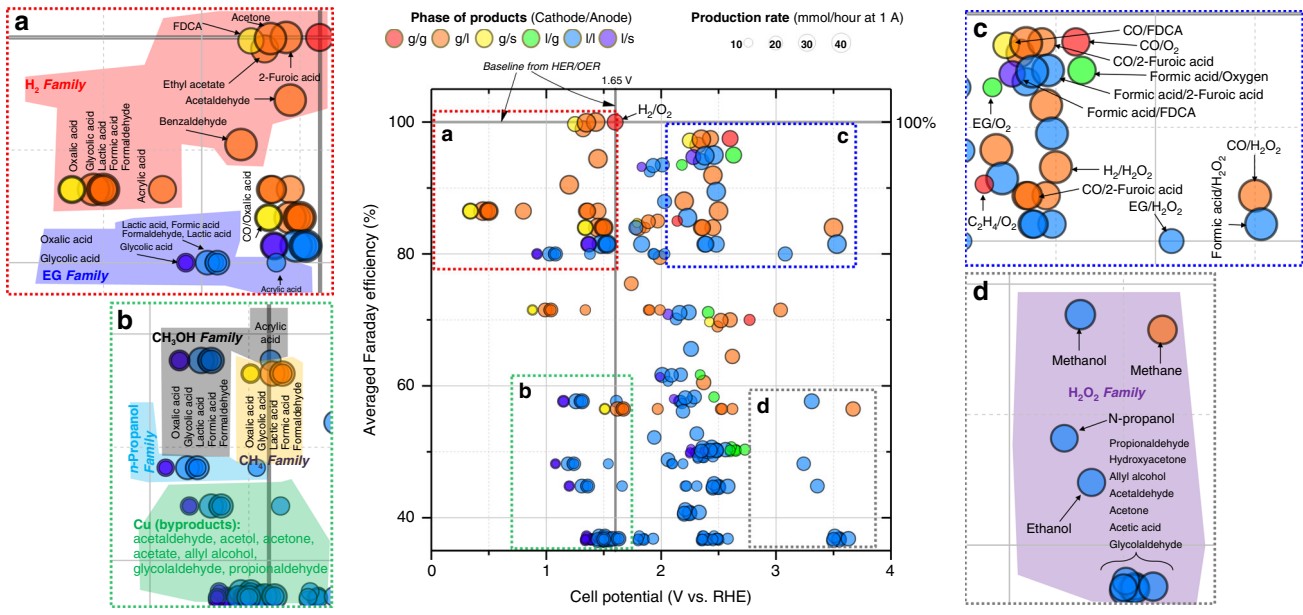

**Fig. 2** Faraday efficiency vs. cell potential for e-chemical paired electrolysis. **a** High FE and low cell potential. **b** Low FE and low cell potential. **c** High FE and high cell potential. **d** Low FE and high cell potential. Some OOR FE data were missing; the lowest value was assumed, 73% for lactic acid from glycerol

**Table 1 The typical characteristics of the main redox reactions in the paired electrolysis system**

| | Product | Characteristics |
|---|---|---|
| *Cathodic reactions* | | |
| Group I | H₂, CO, formic acid, ethylene | -High FE (>70%) |
| | | -High current density (>100 mA cm⁻²) |
| Group II | Methanol, ethanol, *n*-propanol | -Liquid type product |
| | | -Low FE (4 < x < 50 %) |
| | | -Low current density (0.7 < x < 5 mA cm⁻²) |
| | | -Laboratory scale |
| Group III | Methane, acetaldehyde, glyoxal, hydroxyacetone, acetone, acetate, ally alcohol, glycolaldehyde, propionaldehyde, ethylene glycol[a] | -Lowest FE (<1%) |
| | | -Lowest current density (<0.5 mA cm⁻²) |
| | | -Laboratory scale |
| *Anodic reactions* | | |
| Group IV | Acetaldehyde, acetic acid, ethyl acetate, acrylic acid, lactic acid, formaldehyde, formic acid, glycolic acid, oxalic acid | -Cathode linked materials |
| Group V | 2-Furoic acid, 2,5-furandicarboxylic acid | -Biomass intermediates |
| Group VI | Oxygen, hydrogen peroxide, benzaldehyde, benzoic acid, 4-methoxybenzaldehyde, acetophenone, acetone, phenoxyacetic acid | -Other substances |

[a]Ethylene glycol with ionic liquid: 0.3 mA cm⁻², FE 87%[52]

converted into target anode products through an oxidation reaction. All products except oxygen exist in the liquid phase obtained from gas/liquid separation. Thus, the proper liquid/liquid separation process must be carried out to separate the electrolyte solution, the unreacted raw material, and the products.

To ensure analytical reliability, detailed market survey on raw materials, products, and utilities were conducted (Supplementary Tables 1–5). The sizing of unit processes including electrolyzer systems, separation processes, and recycling systems was thoroughly considered by process model, which are employed to estimate the capital cost. A figure of merit (Fig. 2) shows how the performance in terms of current density and FE varies to some extent, but there is a lack of understanding as to the physical limit and which innovative technologies will cause dramatic changes. Therefore, it is necessary to proceed by assuming that there are no other dramatic technical innovations within *e*-chemical systems beyond the current available data.

Through our detailed summary of previous economic analysis[7–9,19–23] (Supplementary Table 6), here, we employed cash flow analysis[24] based on NPV and LCC, which is the most appropriate economic metrics for nonfuel chemicals. The total capital investment ($C_{TCI}$), including total depreciable capital ($C_{TCD}$), and working capital ($C_{WC}$), was calculated by the process considered in Fig. 3 and capital cost. The production cost exclusive of depreciation ($C_{Excl.\ Dep.}$) included the feedstocks (electrolyte, $CO_2$, and organic raw materials), utilities, labor-related operations, maintenance, operating overhead, property taxes and insurance, and general expenses. Depreciation ($C_D$) was considered with modified accelerated cost recovery system (MACRS) depreciation for a 7-year life, and the estimated plant life was 15 years ($N_{life}$), including a 2-year plant construction period. Net earnings (NE) and annual cash flow (CF) were simply calculated by Eqs. (1) and (2), respectively. The LCC was calculated from the cumulative cash flow, but a sequence of LCC

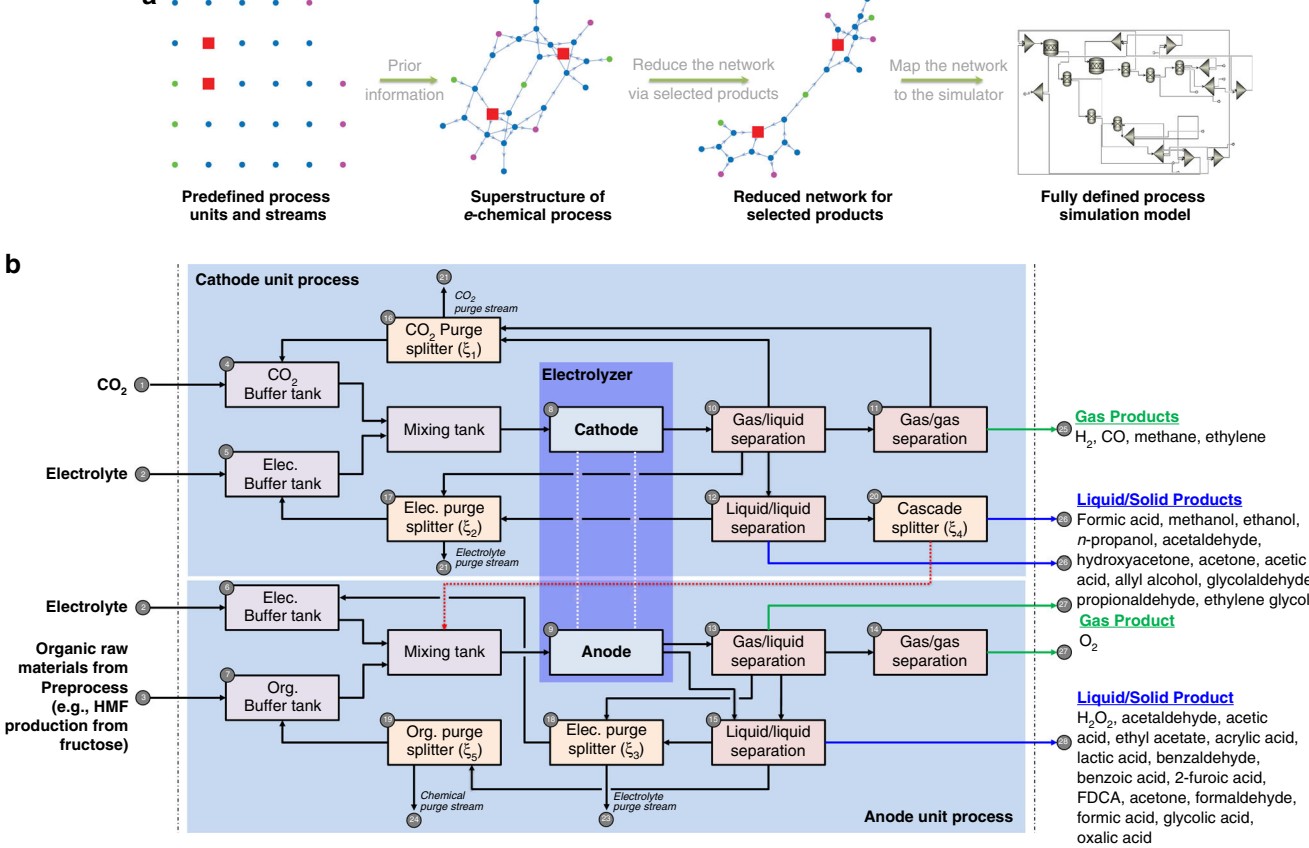

**Fig. 3** Automated and generalized platform for the technoeconomic analysis. **a** Schematic diagram of how the automated platform reduces the process structure and generates the simulation model. **b** Superstructure of the electrochemical coproduction process used to simultaneously consider all possible structures of the process

values were tested until the final NPV reached zero (Eqs. (3) and (4)). The detailed cost information, cash flow sheet, and procedure are given in the Supplementary Methods.

$$\text{Net earnings (NE)} = (S - C_{\text{Excl. Dep.}} - C_{\text{D}}) \cdot (1.0 - t_{\text{income}}), \tag{1}$$

$$\text{Annual cash flow (CF)} = (\text{NE} + C_{\text{D}}) - C_{\text{TDI}}, \tag{2}$$

$$\text{NPV} = \sum_{n=1}^{N_{\text{life}}} \frac{\text{CF}_n}{(1 + \text{i})^n}, \tag{3}$$

$$\text{NPV}(\text{LCC}_{\text{cathode}}, \text{LCC}_{\text{anode}}) = 0 \tag{4}$$

Variance-based global sensitivity analysis (GSA) was performed by Fourier amplitude sensitivity testing (FAST)[25,26] to quantify the effects of current density, FE, and overpotential. Unlike local sensitivity analysis, GSA gives the global index over the sampling region, which quantifies uncertainty more clearly than changing a single variable from a specific base case. We hypothesized that the global sensitivity of each parameter is different for each combination of coproduction processes. Therefore, analyzing the global sensitivity of each coproduction combination can reveal the key factors for an economically feasible process.

The lower and upper bounds of the sampling region are listed in Supplementary Table 7. In particular, the upper bound of current density was determined from the current maximum HER device performance[27]. Since several combinations of CO$_2$RR–OOR can be

galvanic cells (positive Gibbs free energy) in terms of standard reduction potential, the lower bound of overpotential was set to meet 1% of the current cell potential as a positive value, considering only electrolytic cells. Unless the cell potential at zero overpotential was negative, we set the bound of overpotential as 1–100% based on current cell potential values. The overpotential and current density were treated as independent parameters that may be achieved through further catalyst development even if they were actually coupled by the Tafel equation[28] for a specific catalyst.

**Prescreening step**. Due to the arbitrarily large search space of electrochemical coproduction processes and GSA uncertainty parameters, the analysis of CO$_2$RR–OER and HER–OOR is an important prescreening step for identifying the economics of each product. The LCC of CO$_2$RR and OOR can be compared to the industrial market price, which can be used to infer the competitiveness of a technology in current markets. Additionally, this comparison can provide the degree of economy of each technology at various electrochemical technology levels. We conducted GSA for 15 different CO$_2$RR candidates paired with OER and 17 different OOR candidates paired with HER at different electrochemical technology levels regarding the current density, FE at each electrode, and cell overpotential.

Figure 4 illustrates the distribution of LCC and current market price values to identify the competitiveness of technologies for CO$_2$RR–OER and HER–OOR. In general, the LCCs of CO$_2$RR–OER products are higher than the market price. The

**Table 2 The algorithm to reduce superstructure matrix G to sub-structure matrix $\hat{G}$**

| | |
|---|---|
| 1: **Initialize** $\hat{G} = G$. | |
| 2: **Call** pre-defined information: *name, type, and mapping.* | |
| 3: **Define** $P'$, $Q'$, and $CP$, where $p \in P' \subset P$, $q \in Q' \subset Q$. | |
| 4: **if** $phase(x_p^{cathode}) = 'liquid'$, | $^\forall p \in P'$ **then** |
| 5: $\hat{G}_{11,:} \sim= 0$, $\hat{G}_{:,11} \sim= 0.,$ . | -delete gas/gas separation at cathode unit |
| 6: $\hat{G}_{25,:} \sim= 0$, $\hat{G}_{:,25} \sim= 0$. | -delete gas products outlet |
| 7: $\hat{G}_{10,17} \sim= 0$. | -break direct connection liquid mixture to elec. tank |
| 8: **else if** $phase(x_p^{cathode}) = 'gas'$ | $^\forall p \in P'$ **then** |
| 9: $\hat{G}_{12,:} \sim= 0$, $\hat{G}_{:,12} \sim= 0$. | -delete liquid/liquid separation at cathode unit |
| 10: $\hat{G}_{20,:} \sim= 0$, $\hat{G}_{:,20} \sim= 0$. | -deactivate cascade production |
| 11: $\hat{G}_{26,:} \sim= 0$, $\hat{G}_{:,26} \sim= 0$. | -delete liquid products outlet |
| 12: $\hat{G}_{10,16} \sim= 0$. | -$CO_2$/gas products mixture needs gas/gas separation |
| 13: **else** | |
| 14: $\hat{G}_{10,17} \sim= 0$. | -break direct connection liquid mixture-elec. tank |
| 15: $\hat{G}_{10,16} \sim= 0$. | -$CO_2$/gas products mixture needs gas/gas separation |
| 16: **end if** | |
| 17: **if** $phase(x_q^{anode}) = 'liquid'$, | $^\forall q \in Q'$ **then** |
| 18: $\hat{G}_{13,:} \sim= 0$, $\hat{G}_{:,13} \sim= 0$. | -delete gas/liquid separation |
| 19: $\hat{G}_{14,:} \sim= 0$, $\hat{G}_{:,14} \sim= 0..$ | -delete gas/gas separation at anode unit |
| 20: $\hat{G}_{27,:} \sim= 0$, $\hat{G}_{:,27} \sim= 0$. | -delete gas products outlet |
| 21: **else if** $phase(x_q^{anode}) = 'gas'$, | $^\forall q \in Q'$ **then** |
| 22: $\hat{G}_{14,:} \sim= 0$, $\hat{G}_{:,14} \sim= 0$. | -delete gas/gas separation at anode unit |
| 23: $\hat{G}_{15,:} \sim= 0$, $\hat{G}_{:,15} \sim= 0$. | -delete liquid/liquid separation at anode unit |
| 24: $\hat{G}_{28,:} \sim= 0$, $\hat{G}_{:,28} \sim= 0$. | -delete liquid products outlet |
| 25: $\hat{G}_{19,:} \sim= 0$, $\hat{G}_{:,19} \sim= 0$. | -delete organics oxidation related unit |
| 26: $\hat{G}_{3,:} \sim= 0$, $\hat{G}_{:,3} \sim= 0$. | -delete organics oxidation related unit |
| 27: $\hat{G}_{7,:} \sim= 0$, $\hat{G}_{:,7} \sim= 0$ | -delete organics oxidation related unit |
| 28: **else** | |
| 29: $\hat{G}_{14,:} \sim= 0$, $\hat{G}_{:,14} \sim= 0$. | -delete gas/gas separation at anode unit |
| 30: $\hat{G}_{13,18} \sim= 0$. | -break direct connection liquid mixture to elec. tank |
| 31: $\hat{G}_{9,15} \sim= 0$. | -activate the gas/liquid separation at anode unit |
| 32: **end if** | |
| 33: **if** $CP = true \& CP \cap \{p,q\} \neq \phi$, | $^\forall p \in P'$, $^\forall q \in Q'$, **then** |
| 34: $\hat{G}_{12,26} \sim= 0$. | -activate cascade production |
| 35: **else** | |
| 36: $\hat{G}_{20,:} \sim= 0$, $\hat{G}_{:,20} \sim= 0$. | -deactivate cascade production |
| 37: **end if** | |
| 38: **Map** $\hat{G}$ to Aspen Plus simulator (block nodes, stream nodes, and connectivity) | |

fundamental reasons are (1) the disadvantage of pairing with OER (low oxygen market price and high cell potential); (2) the low solubility of $CO_2$ at 1 bar and 25 °C (33.5 mM), which incurs a large electrolyte solution make-up capacity (~50% of operating expenditures (OPEX), as shown in Fig. 5f); (3) the high expense of separating low-concentration liquid products from a large amount of electrolyte solution; and (4) high electricity costs because of the relatively large cell potential compared to that in HER–OER and the high solar PV electricity cost. In particular, products with a low required electrons per unit molecular weight (ethylene: 0.024 mg C$^{-1}$, ethanol: 0.040 mg C$^{-1}$, acetaldehyde: 0.046 mg C$^{-1}$, and propionaldehyde: 0.038 mg C$^{-1}$; mean value: 0.063 mg C$^{-1}$) are affected more. Volatility is another key factor; highly volatile liquid products such as aldehyde and acetone are difficult to recover through vapor–liquid separation. A large portion of the product exists in the vapor phase due to vapor–liquid equilibrium even when a two-stage cold trap flash drum is applied (Fig. 5f). Therefore, the LCC of highly volatile products is higher than other LCCs because of low recovery.

However, compared to photoelectrochemical (PEC) and photovoltaic-electrolytic (PV-E) solar-hydrogen processes, $CO_2$RR–OER products seem to be promising alternatives as their LCC is within the range of values of the levelized cost of hydrogen (LCH) as a product (Fig. 4, light gray band). In addition, solar hydrogen, which has a higher LCH ($5.5–12.1 kg

$^{-1}$) than the commercial process ($1.39 kg$^{-1}$), is generally accepted as a future clean and renewable energy source[22]. Furthermore, if we assume technical developments enabling inexpensive electrolyzers, lower renewable energy costs, and higher electrolyte recycle ratios—parameters that are fixed in this study—most $CO_2$RR–OER products have economic potential even compared to current market prices (Supplementary Fig. 4). Thus, $CO_2$RR–OER has great potential for producing clean fuel and chemicals in the near future despite its current high LCH.

In contrast, the market price of HER–OOR products is within the LCC distribution determined from GSA, showing the feasibility of the technology. HER–OOR does not require a large flow rate of electrolyte to dissolve $CO_2$ and does not need a $CO_2$ separation system, which is essential for $CO_2$RR–OER. Specifically, we monitored the economic feasibility of cathode-linked products (Group IV), biomass intermediates (Group V), and others (Group VI) (Table 1). First, the mean values of LCC for biomass intermediates (2-furoic acid and FDCA) are lower than the current market price. The low market price of furfural and HMF, which are organic raw materials, leads to economic feasibility. OOR products in Group VI do not readily meet feasibility requirements, but the reasons are different for each substance. Benzaldehyde and benzoic acid are economically infeasible due to the high cost of the raw material, benzyl alcohol. In the case of acetone, the number of

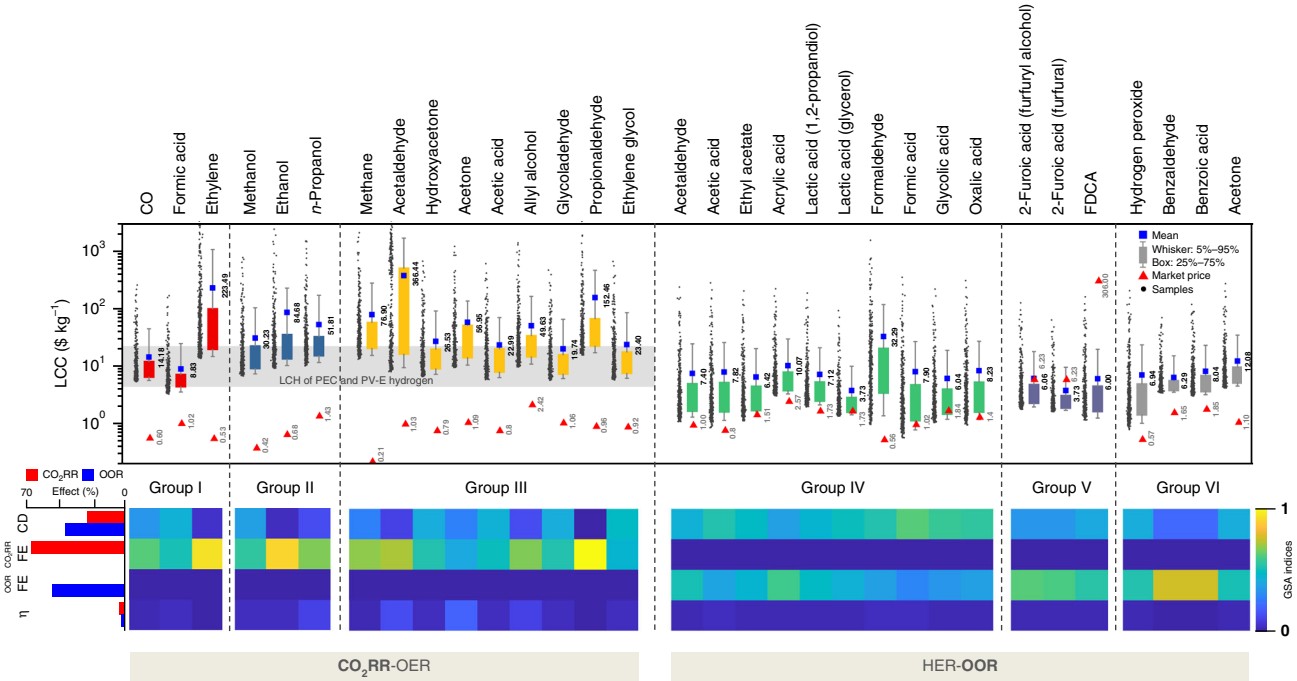

**Fig. 4** Levelized cost of chemicals distribution and global sensitivity analysis. Whole target products of $CO_2RR$–OER and HER–OOR are compared with their market price determined via GSA through current density, FE (cathode and anode), and overpotential. The light gray band represents the levelized cost of hydrogen ($5.5–12.1 kg$^{-1}$) with various types of PEC and PV-E technology[22]

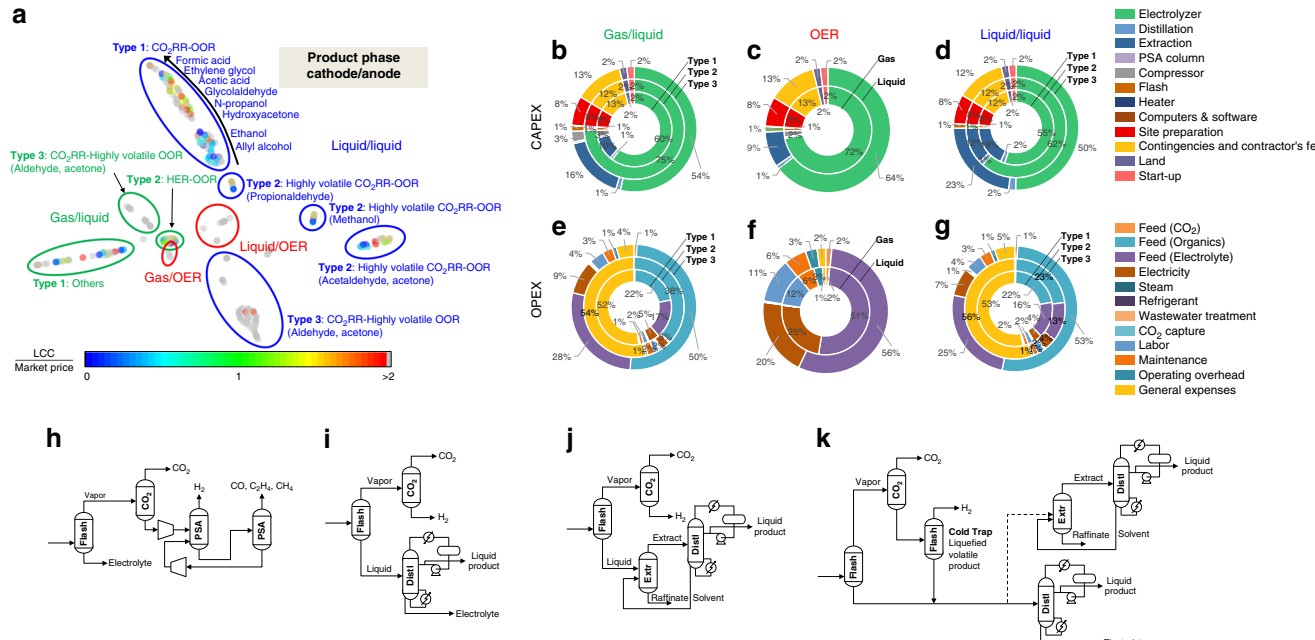

**Fig. 5** All combinations of reduction reactions and oxidation reactions in the base case. **a** Two-dimensional latent space visualization of all the process designs using t-SNE[31]. Positions in the reduced latent space represent similarities in the process structure and mass balance. **b–g** Pie chart visualizing the level of contribution of each component in CAPEX and OPEX to various classes and types clustered in **a**. **h** and **i** Main structural differences in the separation processes for **h** gas products (cathode), **i** light liquid products, **j** heavy liquid or azeotrope products, and **k** highly volatile products

required electrons per unit molecular weight is higher than that of other OOR products, and the productivity is low because of high volatility. Although the raw material for hydrogen peroxide (i.e., water) is inexpensive, the low market price and the high onset potential (2.5 V vs. RHE) lead to a high levelized cost. In Group IV, ethyl acetate, lactic acid, formic acid, glycolic acid, and oxalic acid

can have an LCC lower than the market price. However, OOR products with high volatility (aldehyde) that are generated from expensive raw materials (1,3-propanediol) cannot readily meet the current market price. Altogether, since a considerable number of OOR products are competitive in terms of market price despite process coupling with HER, we can infer that coupling $CO_2RR$ with

OOR is a holy grail in terms of economic feasibility, extensive chemical production portfolios, and carbon capture and utilization.

We also calculated the first-order sensitivity indices from GSA, shown in Fig. 4, which indicate the main effect of the GSA parameters. Unlike HER–OOR, the LCCs of $CO_2RR$–OER products were highly sensitive to FE when the LCC was high. The fundamental reason for the dominant FE sensitivity of the low-economy case is the trade-off relationship between $CO_2RR$ and HER. The current density has a sharp effect on LCC in the very low region ($1\,mA\,cm^{-2}$–$100\,mA\,cm^{-2}$) but exhibits a gradual change in the subsequent region (Supplementary Fig. 5). In general, a high current density has less impact on lowering the LCC over a long range ($100$–$2000\,mA\,cm^{-2}$) for any $CO_2RR$ product. Interestingly, the sign of FE sensitivity depends on the ratio between the market price of $CO_2RR$ products and $H_2$. If the $CO_2RR$–OER process has a deficit in production, the LCC exponentially increases when the FE increases. Because the region at which this sharp transition in LCC occurs differs from that of $CO_2RR$ products, FE sensitivity is decoupled from CD sensitivity for specific products that are less economical. Altogether, relatively economical $CO_2RR$ products (mean LCCs: formic acid: $8.83\,kg^{-1}$, hydroxyacetone: $26.53\,kg^{-1}$, acetic acid: $22.99\,kg^{-1}$, glycolaldehyde: $19.74\,kg^{-1}$, and ethylene glycol: $23.40\,kg^{-1}$) are simultaneously sensitive to current density and FE. However, relatively low-economy $CO_2RR$ products (mean LCCs: ethylene: $223.49\,kg^{-1}$, ethanol: $84.68\,kg^{-1}$, acetaldehyde: $366.44\,kg^{-1}$, and propionaldehyde: $152.46\,kg^{-1}$) are highly sensitive to FE.

Unlike the LCC of $CO_2RR$ products, the LCC of OOR products is sensitive to both current density and FE because the economic competitiveness of OOR products is much higher than that of OER products and the large sales margin of hydrogen from the paired reaction (HER) contributes to the robustness of NPV, even with a low anode FE. Interestingly, FE-sensitive OOR products (acrylic acid, 2-furoic acid, benzaldehyde, and benzoic acid) exist sparsely. The origin of these phenomena is that the market prices of the organic raw materials are relatively expensive (1,3-propanediol: $2.2\,kg^{-1}$, furfural: $1.17$–$1.81\,kg^{-1}$, furfuryl alcohol: $1.25$–$1.87\,kg^{-1}$, and benzyl alcohol: $1.92$–$3.47\,kg^{-1}$), which increases the OPEX contribution of the organic raw material feed to 22–53% (Fig. 5e and g). Therefore, the LCC needs to increase more sharply at low FEs (large volume of nonprofitable oxygen with a low volume of low-profitable OOR products), which boosts the sensitivity of FE.

**Discovering potential products for coproduction processes.** Therefore, we sought to explore all possible combinations of $CO_2RR$–OOR to secure the economic feasibility of the coproduction process and propose various coproduction portfolios for various markets. Figure 5a illustrates the $t$-distributed stochastic neighbor embedding ($t$-SNE)[29] results, representing the 288 $CO_2RR$–OOR electrochemical coproduction processes for which we employed cash flow analysis to perform TEA. All points in the same cluster in the two-dimensional latent space share process structural similarities and stream information. Additionally, we grouped the economic trends of each cluster by marking the LCC-to-market price ratios by color.

First, highly volatile OOR products (gas/liquid type 3 and liquid/liquid type 3) are less economically efficient than when paired with OER. There is a loss from the flash stage to the gas portion, and this loss is difficult to recover despite additional two-stage cold trap flash treatment. Furthermore, the feed cost of organic raw materials increases because of the low recovery of organic raw materials (Fig. 5e and g). In contrast, highly volatile $CO_2RR$ products (liquid/liquid type 2) can achieve economic efficiency when coupled with OOR even though the system has

the same two-stage cold trap flash process. It can be inferred that the economics strongly depend on the OOR product sales, which compensate for the losses of $CO_2RR$. Therefore, for $CO_2RR$–OOR products to achieve a competitive LCC, there should be a sufficiently large recovery in OOR to compensate for shortages in $CO_2RR$ product sales.

In all types in Fig. 5, the electrolyzer bare module cost is ~50–75% of the capital expenditures (CAPEX). Notably, among the total electrolyzer bare module costs, 41% are stack systems and ~60% are catalyst and membrane costs[30]. Therefore, we considered the costs of various catalyst materials for specific reactions to perform a realistic analysis (Supplementary Table 4). After the electrolyzer bare module costs, in descending order, the PSA system (including columns, compressors, and heat exchangers) and distillation system contribute to CAPEX. In the case of OPEX, feeds are overwhelmingly expensive: 13–56% of OPEX is used for electrolyte make-up, even though the electrolyte solution recycle ratio is 90%, and 0–53% is used for organic raw materials. The cost of the feeds can be reduced to some extent depending on how well recycling is performed. The electricity cost is 3–25%, which is relatively high and cannot be reduced by changing the operating conditions. Since OPEX accounts for ~90% of the total production cost, the critical elements in finding potential products for electrochemical coproduction processes are (1) the high recovery of electrochemical products from the electrolyzer, (2) long-term catalyst stability (to reduce maintenance costs), (3) low electrolyte and organic raw material costs, (4) low cell potential (to reduce electric utility costs), and (5) a high mass production rate with a low number of required electrons per unit molecular weight.

We also performed sensitivity analysis regarding the equipment cost of the electrolyzer and separation systems for every $CO_2RR$–OOR electrochemical coproduction processes because our shortcut models may have uncertainly for the real plant application (Supplementary Fig. 6). The flash, distillation, PSA, compressor, and heat exchanger have low impact on LCC sensitivity (<10%) in most cases. The extraction has slightly higher sensitivity but with an average sensitivity of 10%. However, the sensitivity of the electrolyzer can be as high as 100% depending on $CO_2RR$–OOR combination. Interestingly, as the conditions such as FE, current density, overpotential, and electricity cost become lower, the LCC becomes robust to the equipment cost change (Supplementary Fig. 6b). Altogether, the more precise electrolyzer and extraction model are expected to improve the accuracy of LCC, but the current shortcut models can be sufficient for the early stage screening process.

**$CO_2$ reduction and organic oxidation coproduction processes.** We evaluated the performance of $CO_2RR$–OOR electrochemical coproduction processes using GSA across high-performance combinations of the base case (Supplementary Tables 4 and 7) and products of interest regardless of performance (Fig. 6). Every $CO_2RR$ candidate can be economically feasible when paired with Group V products (FDCA and 2-furoic acid) and Group VI products (lactic acid and glycolic acid) in both a wide range and limited range of technology levels (i.e., FE, current density, and overpotential). Ethyl acetate is also economical in a limited range of technology levels when paired with $CO_2RR$ candidates except for gas products (methane, ethylene, and CO). Notably, although FDCA has a very low LCC-to-market price ratio due to its high current market price ($32$–$580\,kg^{-1}$), economic feasibility will be maintained until the market price of FDCA is reduced to $4.25\,kg^{-1}$ at the base case and $1.3\,kg^{-1}$ at the optimal case, regardless of the $CO_2RR$ products (Supplementary Fig. 7).

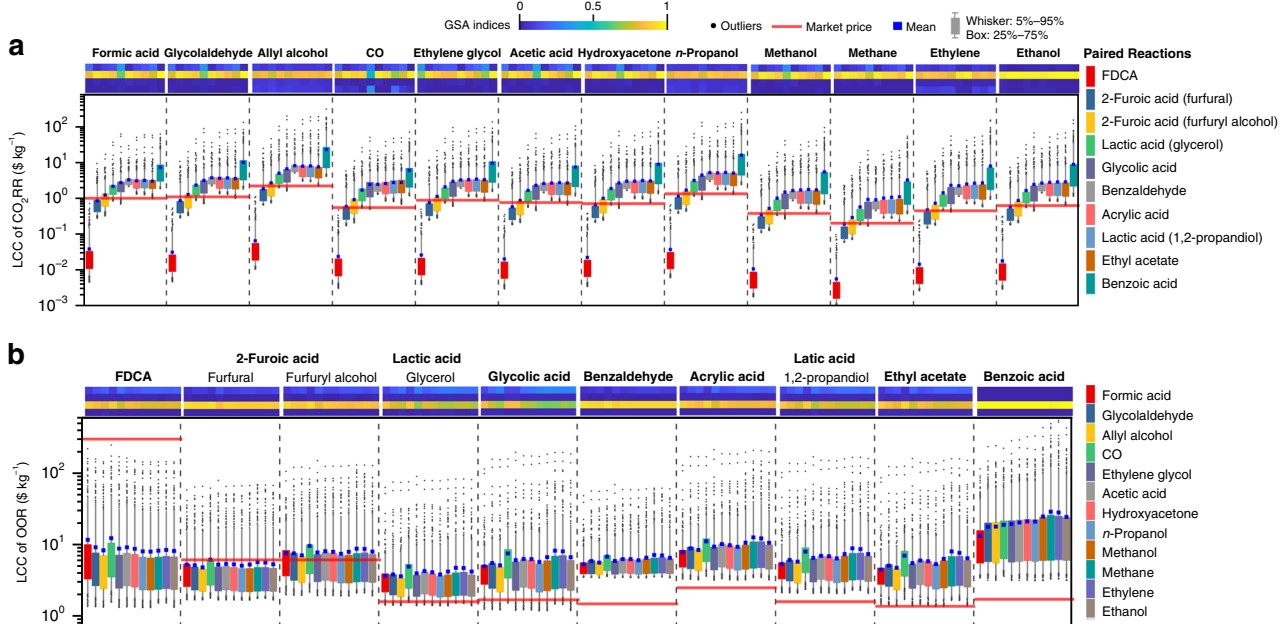

**Fig. 6** Distribution of levelized cost of chemicals with respect to coproduction strategy. **a** LCCs of CO₂RR products paired with 10 different OOR processes. **b** LCCs of OOR products paired with eight different CO₂RR processes

We then explored the GSA of the CO₂RR–OOR system with respect to the current density, FE of each electrode, and overpotential. As OPEX is a greater production cost than CAPEX, FE is generally the key factor. However, in the low-current-density region, the effect of CAPEX becomes very large because a slight change in current density can significantly change the size of the electrolyzer. Similarly, glycolic acid, lactic acid, and ethyl acetate, which have relatively high current density sensitivity, contribute relatively little to the total cost of feed and electricity within the search range. Interestingly, OOR products with high current density sensitivity are economical, and we can develop a design that can minimize OPEX. For a broader perspective, the impact of current density and FE was characterized through a contour plot (Supplementary Fig. 5). As an example, we consider the CO/2-furoic acid (from furfural) electrochemical coproduction process with a detailed explanation in the Supplementary Notes. With a very low technology maturity (CD <10 mA cm⁻² and FE <5%), the LCC-to-market price ratio is dramatically decreased to over 400. Since CD has an inverse relationship with electrolyzer area, CAPEX increases rapidly when CD is lowered. As FE decreases, CAPEX and OPEX do not change much, but LCC sharply increases because the product production rate is proportionally reduced. Economies of scale affect the LCC at specific boundaries that reduce the sensitivity of CD and FE (over 75 mA cm⁻² and 10%, respectively, for CO/2-furoic acid (from furfural)). In summary, at very low performance levels, technological advancements will yield rapid changes, but these effects will decrease after a certain degree of technology maturity is achieved, so it is necessary to check the local sensitivity and reset the most important factor to determine which factor should be improved first.

We evaluated the performance of the cascade structure for 7 CO₂RR–OOR combinations that connect the cathode product stream to the anode organic raw material feed stream to oxidize the cathode product on the anode side (Fig. 4, red dotted line). Although this strategy has the potential to reduce OPEX by supplying organic raw materials for the OOR from CO₂RR products (methanol, ethanol, and ethylene glycol), no meaningful

economic improvements were observed (Supplementary Fig. 8). However, in the case of ethyl acetate from ethanol, glycolic acid from ethylene glycol, and oxalic acid from ethylene glycol, the market price is within the range of leveled cost values, so economic feasibility is possible. Note that commercial glycolic acid production processes often require toxic materials (e.g., formaldehyde, trioxymethylene, and carbon monoxide) and severe operating conditions (e.g., 30 MPa (Du Pont))[31]. Therefore, the cascade coproduction process may be beneficial from an environmental and sustainability point of view as toxic raw materials and high pressure are avoided (Supplementary Table 8). Therefore, when government and industry decision makers consider external factors, such as environmental issues, safety regulation, CO₂ reduction issues, and increased organic raw materials, cascade CO₂RR–OOR could have a relative advantage.

**Optimal case analysis for screening.** Finally, we performed optimal case analysis to determine which CO₂RR–OOR combinations warrant further study and demonstration. The LCC-to-market price ratio for the optimal case was visualized for every combination of CO₂RR–OOR processes (Supplementary Fig. 9, heat map). The elements in Supplementary Table 9 are ordered by the LCC-to-market price ratio and NPV at the end of the plant life. For the CO₂RR, formic acid, n-propanol, acetaldehyde, allyl alcohol, glycolaldehyde, and ethylene glycol are strongly suggested. Several CO₂RR products found in this study have been suggested in previous TEAs[7–9,19,21]. In the case of the OOR, FDCA, 2-furoic acid, ethyl acetate, lactic acid, formic acid, glycolic acid, and oxalic acid are excellent candidates. The worst combinations highly depend on the OOR products and include acetone, formaldehyde, and benzaldehyde, which are ranked similarly to OER products (Supplementary Table 9).

However, CO₂RR products can be promising from the perspective of difficulty of chemical production due to the depletion of fossil fuels. The commercial production processes (current commercial production processes, main applications, and economic aspects for all products are summarized in Supplementary Table 8) of CO, methanol, and ethylene use

syngas, and most other products are based on petrochemical intermediates, such as ethylene, propylene, benzene, and phenol. Therefore, the instability of the petrochemical market and the international trend requiring the dramatic abatement of the use of fossil fuels can trigger the need for technology in the future. However, if possible, the direct synthesis of ethanol, *n*-propanol, and acetaldehyde without synthesizing their precursor, ethylene, will be more advantageous in terms of electrochemistry. Although producing hydrogen peroxide via the OOR is not economically feasible due to the high cell potential and low market price, it cannot be said that there is no future possibility because this approach is an ecofriendly process that can be conducted with only water and without organics, while the current major production processes are organic autoxidation processes, including the anthraquinone process (AO process) and 2-propanol process (Shell process), which require organics[32].

## Discussion

Electrochemical $CO_2RR$–OOR coproduction is a promising route for unconventional chemical production and carbon utilization and can be used as a substitute for conventional petrochemical processes. We developed an automatic TEA platform that generates, calculates, and analyzes process flowsheets without human intervention. This framework was also combined with GSA to decompose the contribution of the LCC of each product into fractions attributed to the current density, FE, and overpotential. We performed 132,768 process calculations to check the economic feasibility across 295 electrochemical coproduction processes, with 16 candidates for the cathode reduction reaction and 18 candidates for the anode oxidation reaction. The full list of potential $CO_2RR$–OOR electrochemical coproduction processes with quantitative economic metrics for proof-of-concept experiments is listed in Supplementary Table 9, and a simplified version is illustrated in Supplementary Fig. 9. These findings provide a wide economic perspective for screening potential $CO_2RR$–OOR candidates and enabling the conceptual design of electrochemical processes.

A limitation of our workflow is that the predefined superstructure does not address various aspects of process systems, such as solidification, acid treatments, and GDL-type electrolyzer devices. This assumption may overlook aspects of formate postprocessing and FDCA crystallization separation, which can alter the production cost. Additionally, GSA was not performed with certain optimal design variables, such as recycle ratio and operating pressure. It is worth to note that the difference in the market size between cathode and anode products make the electrochemical coproduction less attractive, thus their market size also take into account for choosing coproduction pair products. Since the purposes of this study are to accelerate prescreening and conceptual design in a large search space and accomplish sensitivity analysis with a large range of parameters, these issues are acceptable. To confirm the further implications of this study, actual devices and pilot plants using electrochemical $CO_2RR$–OOR need to be empirically demonstrated.

## Methods

**Automatic process model generation**. Since we simultaneously evaluated a wide variety of combinations of processes, it was difficult to manually model them all. There is a need to automate process designs that vary widely depending on the combination of products and process conditions. To build the electrochemical process model for the selected products, we developed an automatic process model generator, which is a key part of the automatic TEA (Supplementary Fig. 2). When we choose the cathode product and the anode product, the automatic process model builder automatically generates a flowsheet as a process simulator file. After inputting the operating conditions, the user can run the simulation to obtain the heat and mass balance.

To generalize the process design, we define the superstructure (which contains all possible combinations of electrochemical processes) and reduce it using algorithm in Table 2 for the chosen products (Fig. 3, Supplementary Figs. 2 and 7). The predefined electrochemical process superstructure matrix $G_{i,j} \in \mathbb{Z}^N \times \mathbb{Z}^N$, $i, j \in U := \{1, \ldots, N\}$ is defined as

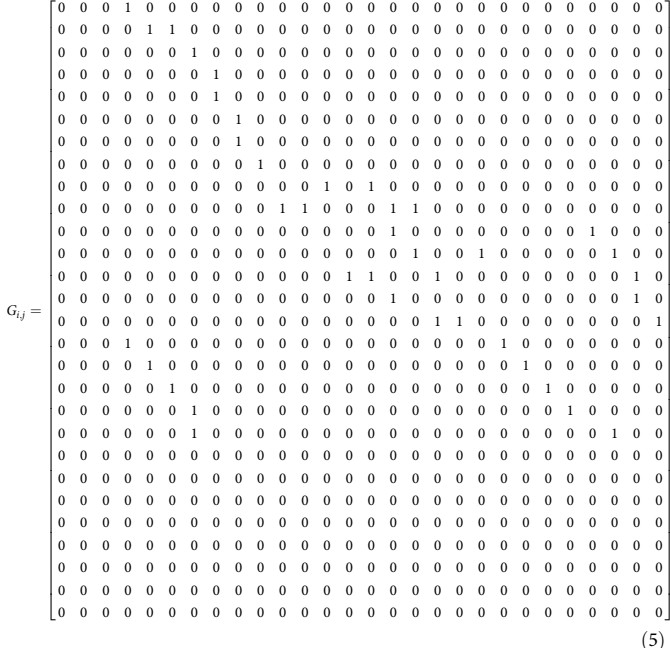

$$(5)$$

where the indices $i$ and $j$ represent nodes (Fig. 3b). Each element $G_{i,j}$ indicates the existence of a connection between the $i$th node and the $j$th node (1: exists and 0: does not exist); $N$ is the total number of nodes; green denotes a simple connection; and magenta denotes a purge stream connection. In this case, $N$ is defined as 28. To mathematically manage the product combinations, we employed $x_p^{cathode} \in \mathbb{Ch}$ and $x_q^{anode} \in \mathbb{Ch}$, corresponding to cathode products and anode products, respectively; these parameters are design variables, where $p \in P \sim = \{1, \ldots, N_c\}$, $q \in Q \sim = \{1, \ldots, N_a\}$, and $\mathbb{Ch}$ is the chemical set. $N_c = 16$ and $N_a = 18$ indicate the number of all kinds of products in the cathode unit and anode unit, respectively. $n_c$ and $n_a$ indicate the number of selected products in the cathode unit and anode unit, respectively. Essential information for the TEA of half-cell reactions, including the number of required electrons ($z$), standard reduction potential, and overpotential (with a literature survey), and market prices of products are arranged in Supplementary Tables 1 and 2.

The proposed process model simulation strategy to obtain information such as the heat and mass balance, energy consumption, and production rate is briefly introduced in Fig. 3. First, the predefined superstructure matrix ($G$) is reduced to the candidate products at the cathode unit ($P'$) and the anode unit ($Q'$). Our automatic process flowsheet generator automatically maps the reduced substructure matrix ($\hat{G}$) under algorithm in Table 2 to the Aspen/Plus simulation file. Then, the user-defined variables, such as the operating conditions (temperature and pressure) and recycle ratio ($\xi_k \in \mathbb{R}$, $k \in K := \{1, \ldots, 5\}$, $\xi_1$: $CO_2$ recycle ratio at the cathode unit, $\xi_2$: electrolyte recycle ratio at the cathode unit, $\xi_3$: electrolyte recycle ratio at the anode unit, $\xi_4$: organic raw material recycle ratio at the anode unit, and $\xi_5$: organic raw material recycle ratio for cascade production), are calculated and added. The generation and calculation pipeline can be efficiently used in TEA for various combinations of products and GSA.

**Energy supply and electrolyzer system**. We assumed that a proton exchange membrane (PEM)-type electrolyzer could be applied in this research. The cost of the electrolyzer was determined from the 2014 DOE hydrogen and fuel cells program[30]. Several studies converted the electrolyzer cost according to specific energy ($\$ kWh^{-1}$) and area ($\$ m^{-2}$) because the corresponding TEAs were performed with a fixed energy supply and varying area via efficiency[8,22,24]. We converted the energy-based electrolyzer cost to an area-based cost to reflect the energy supply from a 40 MW PV farm (solar capacity factor: 20%, PV efficiency: 17%). Since the catalyst metal depends on the chemicals to be produced, additional work was undertaken to reflect this price. Among the total electrolyzer costs, 41% are stack systems and ~60% are catalyst and membrane prices[30]. Therefore, this price was adjusted according to the catalyst metal market price ratio (Supplementary

Table 4).

$$C_{\text{electrolyzerperArea}} = \left( C_{\text{stack}} \times \left( 0.4 + 0.6 \times \frac{C_{\text{catalyst}_{\text{cathode}}} + C_{\text{catalyst}_{\text{anode}}}}{C_{\text{Pt}} + C_{\text{Ir}}} \right) + C_{\text{BoP}} \right)$$
$$\times \text{Installation Factor} \times E \times \text{current density}$$

To calculate the energy and mass balances in the process model, the RStoic reactor model in the Aspen Plus® (Aspen Tech. Inc., Cambridge, MA, USA) and Eqs. (1)–(13) were coupled. If the extent of the reactions is defined by the photovoltaic power plant-/electrolyzer-related equations, then the RStoic reactor model automatically moves the reactions forward and calculates the thermodynamic equilibrium, heat of reactions, and physicochemical properties from the process simulator (Supplementary Fig. 2, calculation stage). From this calculation, the solar-to-chemical efficiency (STC), production rate at the cathode unit ($\dot{n}_p^{\text{out}}$), production rate at the anode unit ($\dot{n}_q^{\text{out}}$), conversion of $CO_2$ ($X_{CO_2}$), required photovoltaic cell area ($A_{\text{PV}}$), and required electrolyzer cell area ($A_{\text{cell}}$) can also be obtained.

The average energy transported from the photovoltaic power station to the electrolyzer plant is calculated by multiplying the capacity of the photovoltaic power station ($E_{\text{farm}}$) and the solar capacity factor ($\eta_{\text{CF}}$) (Eq. (6)). $E_{\text{farm}}$ is predefined to allow consistent economic analysis. Since this value is defined by the maximum possible electrical energy output over a given period, the actual amount of produced electricity considering the influence of daylight, clouds, smog, etc., is lumped into the solar capacity factor ($\eta_{\text{CF}}$).

$$\bar{E}_{\text{PV}} = E_{\text{farm}}\eta_{\text{CF}}. \tag{6}$$

The average required amount of solar energy is defined as

$$\bar{E}_{\text{solar}} = \frac{\bar{E}_{\text{PV}}}{\eta_{\text{PV}}}, \tag{7}$$

where $\eta_{\text{PV}}$ indicates the photovoltaic efficiency. Thus, the total required photovoltaic area at the photovoltaic power station $A_{\text{PV}}$ can be calculated by

$$A_{\text{PV}} = \frac{\bar{E}_{\text{solar}}}{\bar{Q}_{\text{solar}}}, \tag{8}$$

where $\bar{Q}_{\text{solar}}$ represents the average amount of solar energy per unit area in a specific area. The production rates of the selected cathode products and anode products ($\dot{n}_{\text{cathode}}^{\text{out}}$ and $\dot{n}_{\text{anode}}^{\text{out}}$, respectively) are expressed by dividing the partial current density by the number of required electrons for the unit reaction (Eqs. (9) and (10)). In the electrolyzer, the energy used per unit area can be expressed as the product of the full-cell applied potential ($E_{\text{anode}}^0 + \eta_{\text{anode}} - E_{\text{cathode}}^0 + \eta_{\text{cathode}}$) and the current density across the cell, so the electrolyzer area required to consume all the energy transferred from the photovoltaic power station ($\bar{E}_{\text{PV}}$) can be calculated from Eq. (11).

$$\frac{\dot{n}_{\text{cathode}}^{\text{out}}}{A_{\text{cell}}} = \frac{\text{Current density} \cdot \text{FE}_{\text{cathode}}}{F \cdot z_{\text{cathode}}}, \tag{9}$$

$$\frac{\dot{n}_{\text{anode}}^{\text{out}}}{A_{\text{cell}}} = \frac{\text{Current density} \cdot \text{FE}_{\text{anode}}}{F \cdot z_{\text{anode}}}, \tag{10}$$

$$A_{\text{cell}} = \frac{\bar{E}_{\text{PV}}}{\left( E_{\text{anode}}^0 + \eta_{\text{anode}} - E_{\text{cathode}}^0 + \eta_{\text{cathode}} \right) \cdot \text{Current density}}, \tag{11}$$

where $E^0$ and $\eta$ stand for the standard reduction potential and overpotential, respectively. To quantify the performance of the e-chemical process, the unit $CO_2$ conversion at the electrolyzer ($X_{CO_2}$) and solar-to-chemical efficiency (STC) for each product can be defined as

$$X_{CO_2} = \frac{\sum^{\dot{n}_{\text{cathode}}} \cdot c_{\text{cathode}}}{\dot{n}_{CO_2}^{\text{in}}}, \tag{12}$$

$$\text{STC}_x = \frac{E_x^0 \cdot \text{CD}_x \cdot \text{FE}_x \cdot A_{\text{cell}}}{E_{\text{solar}}}, \forall x \in \mathbb{Ch}, \tag{13}$$

where $c_{\text{cathode}}$ is the number of carbon atoms contained in the selected cathode product molecule and the subscript $x$ indicates an arbitrary product in the overall chemical set $\mathbb{Ch}$.

**Separation system.** Herein, we included a target product-oriented separation process for the complete assessment of the system. Both the cathode and anode products inevitably include large amounts of water, unreacted raw materials, and byproducts (e.g., hydrogen and oxygen). A universal separation process applicable to all target products is not available because each target product has different physical and chemical properties. Thus, we designed product-specific separation processes according to the product properties.

The cathode products can be classified according to their normal boiling points and azeotropes. When only hydrogen is generated from the electrolyzer, only the

$CO_2$ capture process is required to recover a high-purity product. In contrast, a pressure swing adsorption process is performed after the $CO_2$ capture process for heavier gas products (i.e., CO, ethylene, and methane) because the gas product stream is a mixture of a target product, $CO_2$, and a hydrogen byproduct. The light liquid products with a normal boiling point lower than that of water are separated using a distillation column. In this case, the liquid stream from the gas/liquid separator consists of water, electrolyte, and products; thus, the light liquid products are recovered as a distillate. The heavy liquids with higher boiling points than that of water cannot be separated in the same manner. If a mixture of water and heavy liquid products is separated through a single distillation column, the water-lean product is recovered as a bottom product, causing the electrolyte to be condensed in the product stream. The electrolyte-containing product stream can be processed to recover high-purity products and recycle electrolytes, but this approach is not desirable because of not only technical difficulties but also economic feasibility. Alternatively, heavy products can be extracted using a solvent and then separated in a distillation column. In this way, water and electrolyte can be recycled without further treatment, and high-purity product recovery is also possible. Several liquid target products have high vapor pressure, and significant amounts of these products (e.g., methanol, acetaldehyde, and acetone) are present in the vapor stream of the gas/liquid separator. In this case, the $CO_2$-lean stream from the $CO_2$ capture process is cooled further and flashed again. Both the capital cost and operating cost are increased as refrigeration and additional gas/liquid separation are introduced, but the product recovery can be greatly improved by the two-stage cold trap/flash process. The liquid product stream is then fed to the product distillation column to recover a high-purity product.

All anode products except oxygen are assumed to be in the liquid phase. Consequently, the anode product stream of the electrolyzer is first separated in a gas/liquid separator to remove the oxygen. The liquid product stream is then extracted using an organic solvent. Note that not only the electrolyte solution and OOR products but also the organic raw materials must be separated. Furthermore, the electrolyte solution and organic raw materials should be recycled to meet economic feasibility requirements. Thus, we assume that all OOR products are first extracted and then separated in a distillation column (Fig. 5j, k). In addition, some light liquid products form an azeotrope with water, which makes an extraction process necessary.

The thermodynamic behavior of separation processes is predicted using the Peng–Robinson equation of state, and properties that are not available in databanks such as those maintained by the National Institute of Standards and Technology (NIST) are estimated using UNIQUAC Functional-group Activity Coefficients (UNIFAC). Most unit operations are modeled using the shortcut method. Shortcut methods are valuable tools for the comprehensive evaluation of key performance indicators in the early phase of conceptual process design[33]. Although the computation of shortcut methods is generally much less expensive than the use of rigorous models, shortcuts can frequently provide essential information required at the conceptual design level[34]. We described the distillation column using Edmister's[35] method, under the assumption of constant relative volatility and molar flow. The dynamic PSA process was simplified by using a shortcut PSA model developed in this study. The PSA shortcut formula was derived based on the Langmuir−Freundlich isotherm, and ideal adsorption and desorption were assumed[36,37]. In the extraction process, we assumed 90% of the products were recovered in the extract stream using methyl tert-butyl ether (MTBE).

The distillation process for separating liquid products from the electrolyte is described by using Edmister's shortcut method[35]. In this approach, the absorption factor and the stripping factor can be found from a relatively simple mass balance once the number of column stages is specified. We estimated the size and cost of a distillation column using a vertical pressurized tray column. Detailed information can be found in the literature[24,38].

We employed linear separators to describe the extraction process. A product-oriented rigorous extraction process is desirable, but the design of such systems is particularly challenging because of the large number of process alternatives and available solvents. We instead assumed 95% product recovery using MTBE. The distillation column for separating the product-rich solvent was modeled using the Edmister shortcut method. We assumed that 90% of the product in the product-rich solvent is recovered in the distillation column. The size and cost of the extraction column and the following solvent product separator can be determined according to the mass and energy balance. The capital cost of the extraction column was calculated by assuming rotating-disk contactor (RDC) liquid–liquid extraction with a maximum throughput of 120 ft$^3$ of liquid per h$^1$ ft$^2$ of column cross-sectional area[24].

Herein, we developed a shortcut pressure swing adsorption (PSA) model using Aspen Custom Modeler. The basic steps of the PSA process consist of adsorption, depressurization, desorption, and pressurization. We developed a mathematical model for each step of the PSA process under the assumption of adsorption equilibrium in batch mode; thus, the early-stage design of the PSA process can be carried out with relatively inexpensive computation. The amount of a component in the adsorbent phases at equilibrium ($q$) can be described by using the Langmuir–Freundlich isotherm model

$$q = \frac{q_{\text{m}}(\text{BP})^{\frac{1}{n}}}{1 + (\text{BP})^{\frac{1}{n}}} \tag{14}$$

where the affinity constant ($B$), saturation capacity ($q_{\text{m}}$), and exponent ($n$) are isotherm parameters and can be expressed in terms of temperature. P is operating

pressure. The adsorption and desorption models were built using Aspen Custom Modeler and integrated into Aspen Plus as model libraries. Therefore, the automatic process model generator can integrate the unit operations (i.e., adsorption, desorption columns, and pressure changers) of PSA into the target-oriented flowsheet. We assumed that the components of the PSA columns are the pressure vessel and zeolite LiX. The capital cost of each unit process was evaluated using Guthrie's method[39]. Detailed equations for the estimation of isotherm parameters and coefficients can be found in Park et al.[40].

**Feedstocks.** The proposed e-chemical process requires three feedstocks: $CO_2$, electrolyte solution, and organic raw materials to be oxidized. To prepare each pure feedstock, the carbon capture process, the electrolyte/water mixing process, and the biomass pretreatment process for organic chemicals (e.g. HMF production from fructose) should be included. Fortunately, several studies have provided estimated or actual production costs via experiments or simulations under various conditions. In this study, we used the estimated costs of feedstocks as determined through the following brief review.

It is necessary to interpret the $CO_2$ capture cost based on the postcombustion $CO_2$ capture process, which is a mature technology that can be implemented in real processes without significant technological developments. In particular, the National Energy Technology Laboratory (NETL) claims that in the monoethanolamine (MEA)-based $CO_2$ capture process (550 MWe subcritical pulverized coal power plant), the capture cost can be as high as $60\ t_{CO_2}^{-1}$, and we used this value for the base case[41]. Although state-of-the-art water-lean solvents have been reported to significantly reduce capture cost[36] (e.g., aminosilicones (GAP-1/TEG) at $50\ t_{CO_2}^{-1}$ [37,42] and nonaqueous solvent-3 (NAS-3) at $47\ t_{CO_2}^{-1}$ [43,44]), such usage is typically in the early development stages; thus, comprehensive assessment and pilot plant testing are still required. A comprehensive review can be found in Heldebrant et al.[36].

The products in Groups IV and VI do not require a specific preprocess to supply organic raw materials, such as alcohol and aldehyde, which are common chemicals in the industry (Table 1 and Supplementary Table 5). Thus, we used the market prices of these two chemicals to estimate feedstock costs. In the case of group V, the organic raw materials are biomass-based products whose supply needs to be confirmed. Industrial furfural production generally consists of the release of pentose by the hydrolysis of lignocellulosic biomass and the cyclodehydration of pentose with a fixed bed reactor and continuous azeotropic distillation using feedstocks such as sugarcane and bagasse/corncobs[45]. In recent years, continuous fractionation has been used with wheat straw or other straws under high temperature and pressure[46]. Furthermore, a multiturbine column (MTC) has been used to achieve a high furfural yield (>80%) in a single-step continuous process[47]. The average market prices of furfural and furfural alcohol are $1.17–1.81 kg$^{-1}$ and $1.25–1.87 kg$^{-1}$, respectively[48,49]. Recently, the "world's first industrial plant" of 99.9% pure HMF with a production rate of 20 t yr$^{-1}$ at AVA Biochem BSL AG was reported[50]. The plant produces HMF by treating energy crops such as wood through modified hydrothermal carbonization (HTC) technology. Although directly obtaining the HMF price from this technique is not possible, a TEA of the HMF production process using a biphasic (aqueous and organic phase) continuously stirred tank reactor (CSTR) reported that the minimum selling price of HMF is $1.33 L$^{-1}$[51]. The electrolyte solution consists of process water and electrolyte, and the prices are $0.2 m$^{-3}$ and $1.38 kg$^{-1}$, respectively. We assumed that 0.1 M $KHCO_3$ was used to operate the process.

## Data availability
The database generated in this study are available at https://www.kist-cepl.com. The additional data that support the findings in this study are available upon reasonable request to the corresponding authors.

## Code availability
The code for the modeling available at https://github.com/ceplkist/Nature-Comm.-TEA-Code.

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

## Acknowledgements

This work was supported by the Korea Institute of Science and Technology (KIST) institutional program and supported by "Next Generation Carbon Upcycling Project" (Project No. 2017M1A2A2046713) through the National Research Foundation (NRF) funded by the Ministry of Science and ICT, Republic of Korea

## Author contributions

J.N., B.S., C.W.L., H.-S.O., and U.L. generalized the electrochemical coproduction concept. J. N. and U.L. developed most of an automated technoeconomic analysis framework and process simulations. J.N. and J.K. did global sensitivity analysis. J.N., B.S., H.J.L., Y.J.H., B.K. M., D.K.L., and H.S.O. investigated the electrochemical properties of each coproduction candidate. J.K. and C.W.L. helped in characterizing market trend analysis. H.-S.O. and U.L. conceived and supervised the project. J.N., B.S., C.W.L., H.-S.O., and U.L. wrote and edited the paper. All authors discussed and commented on the manuscript.

## Competing interests

The authors declare no competing interests.

## Additional information

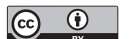

