## [Peer Review File · Nature Communications]

Reviewers' comments:

Reviewer #1 (Remarks to the Author):

This manuscript developed a process synthesis framework and performed extensive techno-economic analysis for a wide range of electrochemical CO₂RR and OOR technologies and processes. The results provide a good reference for screening the economically viable CO₂RR-OOR coproduction processes and will be of interest to the research community in the electrochemical conversion of CO₂. The overall language, figures, tables, etc. in the manuscript are adequate. It can be accepted for publication after addressing the below points:

- 1) The author should be careful to select the market prices for chemicals. For example, the bulk price for FDCA should not be as high as \$32-580/kg. Ref [117] may mislead its market price because of very small quantity sales. FDCA's bulk market price is expected to be competitive to its alternative (e.g., terephthalic acid, ~\$1.5/kg).
- 2) It's good to see downstream separation and recovery are considered, but the simplified separation system with shortcut models may not reflect the real difficulties and costs for separations. Can the authors comment on this point and it would be good to see some sensitivity analysis regarding the key parameters in the separation systems.

Reviewer #2 (Remarks to the Author):

In the present work, the possibilities and limitations for coupling the cathodic CO₂ reduction with the anodic oxidation of organic compounds is explored from the techno-economic point of view. Both half-reaction types are currently studied independently by large scientific communities and chemical companies, whereas a successful experimental coupling of the two reaction types in a single electrochemical cell has so far rarely been achieved. I absolutely agree with the authors that this technology would be a significant advance in sustainable chemicals production, and that the opportunities re. energy efficiency and production cost are immense.

With respect to the (electro)chemistry, the following points need to be addressed:

1. Introduction and conclusion: I miss two very important points in the discussion. A) For parallel processes, divided cells have to be used and electrolysis conditions have to be found, which suit both half-reactions (solvent, salt, pH, temperature ...). The process development is therefore more challenging and the cost significantly higher. B) Naturally, there is a difference in market scale for the products from the anodic and cathodic half-reaction. The bigger the difference, the less attractive is the coproduction.
2. I23: Cement production is another excellent point source of highly concentrated CO₂. Assuming that at some point in the future, all fossil fuels are replaced by renewables, cement production would constitute the largest source of concentrated CO₂. It should therefore also be mentioned here.
3. I64-66: From the perspective of an electrochemist, the explanation sounds a little bit awkward. The anodic and cathodic half reactions always take place simultaneously. They are mutually dependent. According to the definition used here, every existing electrolysis would constitute an "electrochemical coproduction". A better definition would include all electrolytic processes, where both anodic and cathodic half-reactions are used for synthetic purposes.
4. P5, I80 "... the overall theoretical yield is 200%": This is not true. The product yield can never be >100%. It is the FE which reach be up to 200%, since the electric current running through the circuit is used twice.
5. Supporting Information, I38: It is not clear how the generalization that OOR reduces the cell

voltage to 1.44 V was derived. OOR includes a variety of organic compounds with different thermodynamic potentials and overpotentials. The potential at which OOR proceeds can vary within a range of approx. 1.5 V.

6. The authors may be interested in the following paper: J. Am. Chem. Soc. 2016, 138, 46, 15110-15113. It contains a small-scale experimental example of a paired CO₂RR-OOR process and highlights some of the experimental challenges of the approach. This work should definitely be cited.

<List of Major Changes>

For Main Text

1. Page 1, we changed the title to "General Technoeconomic Analysis for Electrochemical Coproduction Coupling CO₂ Reduction with Organic Oxidation".
2. Page 3, we revised sentences regarding point sources of CO₂.
3. Page 6, we revised sentences regarding definition of the electrochemical coproduction.
4. Page 6, we indicate challenges of the parallel electrochemical coproduction.
5. Page 18-19, we added a new result from sensitivity analysis regarding the key parameters in the separation systems.
6. Page 19, we revised sentences regarding market prices for chemicals and sensitivity analysis.
7. Page 35, we added code availability section.
8. Page 36, we added new references (reference number: 1, 15)

For Supporting Information

1. Page S1, we changed the title to "General Technoeconomic Analysis for Electrochemical Coproduction Coupling CO₂ Reduction with Organic Oxidation".
2. Page S3, we revised sentence regarding the cell voltage range of HER–OOR coupling.
3. Page S44, we include Supplementary Fig. 7. indicating sensitivity of LCC over market price through variation of FDCA market price.
4. Page S40, we include Supplementary Fig. 6. showing sensitivity analysis of capital cost variations for every CO₂RR–OOR processes.

Please note that the author's responses are written in blue color and any change in the manuscript and support information is highlighted

Responses to the Comments of the Reviewer 1

This manuscript developed a process synthesis framework and performed extensive techno-economic analysis for a wide range of electrochemical CO₂RR and OOR technologies and processes. The results provide a good reference for screening the economically viable CO₂RR-OOR coproduction processes and will be of interest to the research community in the electrochemical conversion of CO₂. The overall language, figures, tables, etc. in the manuscript are adequate. It can be accepted for publication after addressing the below points:

1. (Reviewer's Comment) *The author should be careful to select the market prices for chemicals. For example, the bulk price for FDCA should not be as high as \$32-580/kg. Ref [117] may mislead its market price because of very small quantity sales. FDCA's bulk market price is expected to be competitive to its alternative (e.g., terephthalic acid, ~\$1.5/kg).*

(Authors' Response) Thank you for your valuable comment. We strongly agree that the FDCA market price in Ref [168] (originally Ref [117]) can be overestimated. Data in Ref [168] is based on export between India and other countries, which contains small quantity to large capacity bulk FDCA. As far as we know, the industrial scale FDCA sales information is rarely available. Thus, we referred the Indian export data ranged from \$32/kg to \$580/kg.

In order to respond possibilities that the FDCA market price get cheaper than its alternatives such as terephthalic acid mentioned by reviewers, we performed additional economic analysis. We monitored the LCC over market price ratio via changing FDCA market price as \$0.1-10/kg and identified minimum market price of FDCA that secures economic feasibility, regardless CO₂RR products. The result indicates that economic feasibility will be maintained for all cathode products until the market price of FDCA is reduced to \$4.25/kg at the base case and \$1.3/kg at the optimal case. On the other hand, none of the CO₂RR-FDCA coproduction secures economic feasibility as the market price of FDCA is reduce below \$2.60/kg at the base case and \$0.63/kg at the optimal case. We add this results in the manuscript and provide the related figure in Supplementary Fig. 7.

Changes made:

- In page 19 of manuscript

Original sentence: “Notably, although FDCA has a very low LCC-to-market price ratio due to its high current market price (\$32 – 580 kg⁻¹), economic feasibility will be maintained until the market price of FDCA is reduced to ~\$2 kg⁻¹, regardless of the CO₂RR products.”

Revision: Notably, although FDCA has a very low LCC-to-market price ratio due to its high current market price (\$32 – 580 kg⁻¹), economic feasibility will be maintained until the market price of FDCA is reduced to \$4.25 kg⁻¹ at the base case and \$1.3 kg⁻¹ at the optimal case, regardless of the CO₂RR products (Supplementary Fig. 7).

- In page 44 of supplementary information

Revision:

Supplementary Fig. 7 Sensitivity of LCC over market price through variation of FDCA market price. a, Based on base case parameters. **b,** Based on optimal case II parameters (Supplementary Table 8). When the market price of FDCA is over \$4.25 kg⁻¹ at base case and \$1.30 kg⁻¹ at optimal case, CO₂RR–OOR process can secure the economic feasibility, regardless of the CO₂RR products.

2. (Reviewer's Comment) *It's good to see downstream separation and recovery are considered, but the simplified separation system with shortcut models may not reflect the real difficulties and costs for separations. Can the authors comment on this point and it would be good to see some sensitivity analysis regarding the key parameters in the separation systems.*

(Authors' Response) Thank you for your important comment on the possibility of model-actual plant mismatch. We agree that the shortcut models can underestimate the separation cost and cannot reflect various real difficulties such as flooding, cracking, pipeline blockage, and etc. Although we designed product-specific separation processes according to the product properties, our process design, sizing, and costing still contain high degree of uncertainty. According to your comment, we additionally performed sensitivity analysis on the electrolyzer and separation systems for every of CO₂RR–OOR electrochemical coproduction processes. As a result, we identify the electrolyzer and the extraction process have large influence on LCC and should be improved for more accurate LCC calculation. However, the shortcut models can be sufficient for the screening stage conceptual design¹ and the sensitivity (<10) of the optimal case is not so significant.

In the case of real difficulties, our approach is based on “conceptual design”, which is usually performed when the technology is at low level of technology readiness level (TRL). Most of electrochemical CO₂RR-OOR coproduction technologies are in the relatively low TRL as compared with commercial chemical processes, thus we believe that conceptual design is appropriate to conduct extensive comparative technoeconomic analysis for screening processes. The detail designs based on rigorous process models can be an alternative approach, however, they are not generally adequate for screening process alternatives because of expensive computation, large numbers of parameter, and requirement of experimental data. According to your comment we include methods and results in main manuscript and related figure in Supplementary Fig. 6.

Changes made:

- In page 18-19 of manuscript

Revision: We also performed sensitivity analysis regarding the equipment cost of the electrolyzer and separation systems for every CO₂RR–OOR electrochemical coproduction processes because our shortcut models may have uncertainly for the real plant application (Supplementary Fig. 6). The flash, distillation, PSA, compressor, and heat exchanger have low impact on LCC sensitivity (<10%) in most cases. The extraction has slightly higher sensitivity but with an average sensitivity of 10%. However, the sensitivity of the electrolyzer can be as high as 100% depending on CO₂RR-OOR combination. Interestingly, as the conditions such as FE, current density, overpotential, and electricity cost become lower, the LCC becomes robust to the equipment cost change (Supplementary Fig. 6b). Altogether, the more precise electrolyzer and extraction model are expected to improve the accuracy of LCC, but the current shortcut models can be sufficient for the early stage screening process.

- In page 40 of manuscript

Revision:

Supplementary Fig. 6 Sensitivity analysis of capital cost variations for every CO₂RR-OOR processes. a, Based on base case parameters. **b,** Based on optimal case II parameters (Supplementary Table 9). The definition of sensitivity for a unit *i* is given by,

$$\text{sensitivity}_i = \frac{\partial \left(\frac{\text{LCC}}{\text{Market prices}} \right)}{\partial (\text{ratio}_{\text{equipment cost}})}$$

where $\text{ratio}_{\text{equipment}}$ represents equipment cost changed ratio compared to the equipment cost evaluated by the proposed shortcut electrolyzer and separation models. For example, the meaning of $\text{sensitivity}_{\text{PSA}}$ equals to 0.5 is that if equipment cost of PSA increases by 100%, then LCC increases by 50%.

Responses to the Comments of the Reviewer 2

In the present work, the possibilities and limitations for coupling the cathodic CO₂ reduction with the anodic oxidation of organic compounds is explored from the techno-economic point of view. Both half-reaction types are currently studied independently by large scientific communities and chemical companies, whereas a successful experimental coupling of the two reaction types in a single electrochemical cell has so far rarely been achieved. I absolutely agree with the authors that this technology would be a significant advance in sustainable chemicals production, and that the opportunities re. energy efficiency and production cost are immense.

With respect to the (electro)chemistry, the following points need to be addressed:

1. (Reviewer's Comment) *Introduction and conclusion: I miss two very important points in the discussion. A) For parallel processes, divided cells have to be used and electrolysis conditions have to be found, which suit both half-reactions (solvent, salt, pH, temperature ...). The process development is therefore more challenging and the cost significantly higher. B) Naturally, there is a difference in market scale for the products from the anodic and cathodic half-reaction. The bigger the difference, the less attractive is the coproduction.*

(Authors' Response) Thank you for the reviewer's important comment and authors completely agree with that the parallel processes can be challenging and more expensive. In order to respond your comment, we include Supplementary Table 1 indicating typical operating conditions for cathodic and anodic products. As shown in the table, numbers of CO₂RR-OOR combination can have significantly different operating conditions, and they eventually result higher electrolyzer cost. Therefore, we carried out a sensitivity analysis demonstrating effect of equipment cost change (i.e., flash, distillation, extraction, PSA, compressors, and heat exchangers) on LCC (Supplementary Fig. 6). Please note that the definition of sensitivity for a unit *i* is given by,

$$\text{sensitivity}_i = \frac{\partial \left(\frac{\text{LCC}}{\text{Market prices}} \right)}{\partial (\text{ratio}_{\text{equipment cost}})}$$

where $\text{ratio}_{\text{equipment}}$ represents equipment cost changed ratio compared to the equipment cost evaluated by the proposed shortcut electrolyzer and separation models. For example, the meaning of $\text{sensitivity}_{\text{PSA}}$ equals to 0.5 is that if equipment cost of PSA increases by 100%, then LCC increases by 50%. The result shows that the sensitivity of the electrolyzer can be as high as 100% depending on CO₂RR-OOR combination. Interestingly, as the conditions such as FE, current density, overpotential, and electricity cost become lower, the LCC becomes robust to the equipment cost change (Supplementary Fig. 6b). The maximum sensitivity of the electrolyzer at the optimal case II with 95% confidence is about 10% and it can be interpreted as LCC only changes by 10% although the electrolyzer cost is doubled.

We also strongly agree that the electrochemical coproduction becomes less attractive if a significant differences exist in product market sizes. Thus, we revise the conclusion and

indicate that the difference in the market size between cathode and anode products make the electrochemical coproduction less attractive. In addition, we include market sizes of the chemicals investigated in this study for providing related information to readerships. (Supplementary Table 8)

Changes made:

- In page 6 of manuscript

Original sentence: “The most well-known, industrially established example is the chlor-alkali process, wherein chlorine and sodium hydroxide are produced at the anode and cathode, respectively.¹⁴”

Revision: “The most well-known, industrially established example is the chlor-alkali process, wherein chlorine and sodium hydroxide are produced at the anode and cathode, respectively.¹⁴ Interestingly, small-scale CO₂RR–OOR process demonstration regarding the oxidative condensation through molecular electrocatalysts belongs to a parallel type.¹⁵ The parallel paired electrolysis can be very challenging if significant differences exist between half reaction operating conditions (i.e., solvent, pH, temperature, etc.) and the different operating conditions may cause expensive electrolyzer design and fabrication cost. We summarize operating conditions of both cathodic and anodic products in Supplementary Table 1.”

- In page 12 of supplementary information

Revision:

Supplementary Table 1 Electrolysis conditions of cathodic/anodic products in the parallel process

Product	pH	Temperature (°C)	Electrolyte	Ref.
Cathode products				
H ₂	0 - 1	25 - 80	H ₂ SO ₄ / HClO ₄ Solution, PEM	16, 72, 73, 74, 75,
	13 - 14	25 - 80	KOH Solution, AEM	76
Syngas	6 - 7	25	KHCO ₃ Solution, AEM	77, 78, 79, 80
CO	6 - 7	25	KHCO ₃ Solution, AEM	81, 82, 83
Formate	6 - 7	25	KHCO ₃ Solution, AEM	84, 85, 86
Methanol	6 - 7	25	KHCO ₃ Solution, AEM	87, 88
Methane	6 - 7	25	KHCO ₃ Solution, AEM	87, 89
Ethylene	6 - 7	25	KHCO ₃ Solution, AEM	90, 91, 92
Ethanol	6 - 7	25	KHCO ₃ Solution, AEM	93, 94, 95
n-Propanol	6 - 7	25	KHCO ₃ Solution, AEM	87, 93, 96
Acetaldehyde	6 - 7	25	KHCO ₃ Solution, AEM	87, 97
Glyoxal	6 - 7	25	KHCO ₃ Solution, AEM	87, 98
Acetone	6 - 7	25	KHCO ₃ Solution, AEM	87, 99
Acetate	6 - 7	25	KHCO ₃ Solution, AEM	87, 100
Ethylene glycol	6 - 7	25	KHCO ₃ Solution, AEM	87, 101
Anode Products				
O ₂	0 - 1	25 - 80	H ₂ SO ₄ / HClO ₄ Solution, PEM	102, 103
	13 - 14	25 - 80	KOH Solution, AEM	
Hydrogen peroxide	0 - 1	25	H ₂ SO ₄ / HClO ₄ Solution, PEM	104, 105
	13 - 14	25	KOH / NaOH Solution	
Acetaldehyde	0 - 1	25 - 80	H ₂ SO ₄ / HClO ₄ Solution	106, 107

Acetic acid	0 - 1	25 - 80	H ₂ SO ₄ / HClO ₄ Solution	106, 107, 108
Ethyl acetate	13 - 14	25	KOH Solution	109
Acrylic acid	0 - 1	25	HClO ₄ Solution	110
Lactic acid	13 - 14	25	KOH Solution, AEM	111
Benzaldehyde	-	45	Ionic Liquid [Bmin][BF ₄]	112, 113
Benzoic acid	0 - 1	25 - 90	H ₂ SO ₄ Solution	114, 115
2-Furoic acid	9.4	25 - 80	NaHCO ₃ /Na ₂ CO ₃	116, 117
	13 - 14	30	KOH Solution	
2,5-Furandicarboxylic acid (FDCA)	1	60	H ₂ SO ₄ Solution	118, 119
	13 - 14	25	KOH Solution	
4-Methoxybenzaldehyde	-	25	TEMPO Solution	120
Acetophenone	-	25	CH ₂ Cl ₂ Solution	121
Acetone	13 - 14	25	KOH Solution	122, 123
Phenoxyacetic acid	13 - 14	25	NaOH Solution	124
Formaldehyde	0 - 1	25	HClO ₄ Solution	125, 126
Formic acid	0 - 1	25	H ₂ SO ₄ Solution	127, 128
Glycolic acid	13 - 14	25	KOH Solution	129
Oxalic acid	13 - 14	25	KOH Solution	129

* PEM: Proton Exchange Membrane, * AEM: Anion Exchange Membrane

- In page 6 of manuscript

Original sentence: “Additionally, GSA was not performed with certain optimal design variables, such as recycle ratio and operating pressure.”

Revision: “Additionally, GSA was not performed with certain optimal design variables, such as recycle ratio and operating pressure. It is worth to note that the difference in the market size between cathode and anode product make the electrochemical coproduction less attractive, thus their market size also take into account for choosing coproduction pair products.”

2. (Reviewer's Comment) *l23: Cement production is another excellent point source of highly concentrated CO₂. Assuming that at some point in the future, all fossil fuels are replaced by renewables, cement production would constitute the largest source of concentrated CO₂. It should therefore also be mentioned here.*

(Authors' Response) Thank you for your valuable comment. We completely agree that cement production is an important CO₂ point source and should not be ignored. Thus, we mentioned cement production in the manuscript according to IEA report²

Changes made:

- In page 19 of manuscript

Original sentence: “Point sources with intensive CO₂ concentrations are located at power plants, cement production, and petrochemical facilities where carbon capture and utilization can be accomplished to create carbon-neutral cycles.¹”

Revision: “Point sources with intensive CO₂ concentrations are located at power plants, **cement production**, and petrochemical facilities where carbon capture and utilization can be accomplished to create carbon-neutral cycles.^{1,2}”

3. (Reviewer's Comment) 164-66: *From the perspective of an electrochemist, the explanation sounds a little bit awkward. The anodic and cathodic half reactions always take place simultaneously. They are mutually dependent. According to the definition used here, every existing electrolysis would constitute an "electrochemical coproduction". A better definition would include all electrolytic processes, where both anodic and cathodic half-reactions are used for synthetic purposes.*

(Authors' Response) Thank you for your comment and we apologize for awkward definition of electrochemical coproduction. For more clear explanation of the electrochemical coproduction that we intended to describe in this manuscript, we modified original sentence as below.

Changes made:

- In page 6 of manuscript

Original sentence: “To run an electrolysis cell, two half-reactions, oxidation and reduction, should be paired to create a complete redox reaction. Such coupling of reduction and oxidation reactions is defined as electrochemical coproduction.”

Revision: “To run an electrolysis cell, two half-reactions, oxidation and reduction, should be paired to create a complete reaction. Herein, we define the electrochemical coproduction as a paired electrolysis that both cathodic CO₂RR and anodic OOR produce chemicals with market values.”

4. (Reviewer's Comment) *P5, l80 "... the overall theoretical yield is 200%." : This is not true. The product yield can never be >100%. It is the FE which reach be up to 200%, since the electric current running through the circuit is used twice.*

(Authors' Response) We apologize for the confusing statement. Per your comment, we removed original sentence for clarification.

- In page 7 of manuscript

Original sentence: "If the current efficiency in each reaction is 100%, the overall theoretical yield is 200%."

Revision: removed

5. (Reviewer's Comment) *Supporting Information, 138: It is not clear how the generalization that OOR reduces the cell voltage to 1.44 V was derived. OOR includes a variety of organic compounds with different thermodynamic potentials and overpotentials. The potential at which OOR proceeds can vary within a range of approx. 1.5 V.*

(Authors' Response) We appreciate and agree with the reviewer's comment that the general cell voltage range of HER–OOR coupling should consider wide range of organic oxidation reactions.

Changes made:

- In page 3 of supplementary information

Original sentence: “Recently, the electrochemical oxidation of organic chemicals has emerged as an effective reaction for coupling with the HER; such coupling significantly reduces the overall cell voltage to approximately 1.44 V, which is 200–300 mV lower than that of HER–OER coupling.”

Revision: “Recently, the electrochemical oxidation of organic chemicals has emerged as an effective reaction for coupling with the HER; such coupling significantly reduces the overall cell voltage to within a range of approximately 1.5 V, which is 100–300 mV lower than that of HER–OER coupling.”

6. **(Reviewer's Comment)** *The authors may be interested in the following paper: J. Am. Chem. Soc. 2016, 138, 46, 15110-15113. It contains a small-scale experimental example of a paired CO₂RR-OOR process and highlights some of the experimental challenges of the approach. This work should definitely be cited.*

(Authors' Response) Thank you for your comment. We totally agree with this is a very important and interesting paper which contains a good example of a paired CO₂RR-OOR process and highlights some challenges. We cited the paper in the manuscript Ref [15] that introduce the concept of a paired CO₂RR-OOR process.

Changes made:

- In page 6 of manuscript

Original sentence: “Parallel paired electrolysis features the simultaneous occurrence of two unrelated half-reactions in a divided cell. The most well-known, industrially established example is the chlor-alkali process, wherein chlorine and sodium hydroxide are produced at the anode and cathode, respectively.¹⁴”

Revision: “Parallel paired electrolysis features the simultaneous occurrence of two unrelated half-reactions in a divided cell. The most well-known, industrially established example is the chlor-alkali process, wherein chlorine and sodium hydroxide are produced at the anode and cathode, respectively.¹⁴ Interestingly, small-scale CO₂RR–OOR process demonstration regarding the oxidative condensation through molecular electrocatalysts belongs to a parallel type.¹⁵”

References

1. Redepenning C, Recker S, Marquardt W. Pinch□based shortcut method for the conceptual design of isothermal extraction columns. *AIChE Journal* **63**, 1236-1245 (2017).
2. Efficiency E. Tracking industrial energy efficiency and CO₂ emissions. *International Energy Agency* **34**, 1-12 (2007).

Reviewers' comments:

Reviewer #1 (Remarks to the Author):

The reviewer's points are appropriately addressed. The paper is now acceptable for publication.

Reviewer #2 (Remarks to the Author):

Apart from comment #5, my concerns and suggestions have been appropriately addressed by the authors. It is important to note that great efforts have been made to deal with the problem of different electrolysis conditions and different market sizes (comment #1).

Regarding comment #5: The changed sentence now reads "Recently, the electrochemical oxidation of organic chemicals has emerged as an effective reaction for coupling with the HER; such coupling significantly reduces the overall cell voltage to within a range of approximately 1.5 V, which is 100–300 mV lower than that of HER–OER coupling".

The authors should once again draw their attention to the consideration of the potential differences between HER and OOR (see page 3 of the SI). First of all, the amended sentence makes no sense (the first half sentence does not fit the second half sentence and the rest of the discussion with regard to the numbers). In addition, however, the fact that the oxidation potentials of organic compounds vary greatly depending on the functional group to be oxidized and on electrolysis conditions (electrolyte, pH, electrode material) is not sufficiently taken into account in the discussion. In my opinion, the originally assumed cell voltage of 1.44 V for HER–OOR coupling was too rough a generalization and should therefore be given with the appropriate range of fluctuation.

<List of Major Changes>

For Main Text

1. Page 35, we added a data availability section.
2. Page 35, we modified the code availability section.
3. Page 46, we modified the caption of **Figure 1b**.

For Supporting Information

1. Page S3, we revised the sentence regarding OER oxidation potential.
2. Page S3, we revised the paragraph regarding potential advantages of OOR coupling.

Please note that the author's responses are written in blue color and any change in the manuscript and support information is highlighted

Responses to the Comments of the Reviewer 2

Apart from comment #5, my concerns and suggestions have been appropriately addressed by the authors. It is important to note that great efforts have been made to deal with the problem of different electrolysis conditions and different market sizes (comment #1).

1. (Reviewer's Comment) *The changed sentence now reads “Recently, the electrochemical oxidation of organic chemicals has emerged as an effective reaction for coupling with the HER; such coupling significantly reduces the overall cell voltage to within a range of approximately 1.5 V, which is 100–300 mV lower than that of HER–OER coupling”.*

The authors should once again draw their attention to the consideration of the potential differences between HER and OOR (see page 3 of the SI). First of all, the amended sentence makes no sense (the first half sentence does not fit the second half sentence and the rest of the discussion with regard to the numbers). In addition, however, the fact that the oxidation potentials of organic compounds vary greatly depending on the functional group to be oxidized and on electrolysis conditions (electrolyte, pH, electrode material) is not sufficiently taken into account in the discussion. In my opinion, the originally assumed cell voltage of 1.44 V for HER–OOR coupling was too rough a generalization and should therefore be given with the appropriate range of fluctuation.

(Authors' Response) Thank you for your valuable comment. We apologize for unclear statement. We agree with the reviewer's concerns regarding the range of cell voltage for the HER–OOR couplings. As the reviewer pointed out, the cell voltage varies depending on the combination of the reactions. In Supplementary Table 3, we include both standard potential and over potential of the oxidation products. According to Supplementary Table 3, several organic oxidation products show lower oxidation potential than oxygen (**Figure 1b**). For example, HMF oxidation reaction has 1.25 V onset potential and can achieve 20 mA/cm² current density at 1.32 V¹. The methanol oxidation reaction to formic acid occurs even lower potential (>0.6 V) than the standard reduction potential of OER². In order to state the potential advantage of ORR we modified original sentence as below.

In addition, the electrolysis conditions (electrolyte, pH, electrode material) could also influence on the cell voltage. To address the concerns raised by the reviewer, we revised the Supplementary Table 1 to show catalyst material, pH, and electrolyte that are generally used for both cathode and anode products.

Changes made:

- In page 3 of supplementary information

Original sentence: “Recently, the electrochemical oxidation of organic chemicals has emerged as an effective reaction for coupling with the HER; such coupling significantly reduces the overall cell voltage to within a range of approximately 1.5 V, which is 100–300 mV lower than that of HER–OER coupling.”

Revision: “Recently, the electrochemical organic oxidation reaction (OOR) has emerged as an effective alternative for coupling with the HER. HER-OOR coupling is an attractive option not only because of economically high value products but because of potentially low overall cell voltage. For example, HMF oxidation reaction has 1.25 V onset potential and can achieve 20 mA/cm² current density at 1.32 V. The methanol oxidation reaction to formic acid occurs even lower potential (> 0.6 V) than the standard reduction potential of oxygen. The overall cell potential adopted in this study for each coupling reaction can be calculated using Supplementary Table 2 and 3. Note that the overall cell voltages of organic compounds vary depending on the functional group to be oxidized and on electrolysis conditions such as electrolyte, pH, and electrode materials. Thus, we analyze the influence of the overall cell potential on the economic potential through global sensitivity analysis.

Revised Supplementary Table 1 Electrolysis conditions of cathodic/anodic products in the parallel process

Product	Catalyst	pH	Temperature (□)	Electrolyte	Ref.
Cathode products					
H ₂	Pt, MoS ₂	0 - 1	25 - 80	H ₂ SO ₄ / HClO ₄ Solution, PEM	16,72-76
	Pt	13 - 14	25 - 80	KOH Solution, AEM	
Syngas	Ag	6 - 7	25	KHCO ₃ Solution, AEM	77-80
CO	Ag	6 - 7	25	KHCO ₃ Solution, AEM	81-83
Formate	Sn, SnO ₂	6 - 7	25	KHCO ₃ Solution, AEM	84-86
Methanol	Cu	6 - 7	25	KHCO ₃ Solution, AEM	87,88
Methane	Cu	6 - 7	25	KHCO ₃ Solution, AEM	87,89
Ethylene	Cu	6 - 7	25	KHCO ₃ Solution, AEM	90-92
Ethanol	Cu, CuAg	6 - 7	25	KHCO ₃ Solution, AEM	93-95
n-Propanol	Cu, CuZnO	6 - 7	25	KHCO ₃ Solution, AEM	87,93,96
Acetaldehyde	Cu	6 - 7	25	KHCO ₃ Solution, AEM	87,97
Glyoxal	Cu	6 - 7	25	KHCO ₃ Solution, AEM	87,98
Acetone	Cu	6 - 7	25	KHCO ₃ Solution, AEM	87,99
Acetate	Cu	6 - 7	25	KHCO ₃ Solution, AEM	87,100
Ethylene glycol	Cu, Au	6 - 7	25	KHCO ₃ Solution, AEM	87,101
Anode Products					
O ₂	Ir, Ru	0 - 1	25 - 80	H ₂ SO ₄ / HClO ₄ Solution, PEM	102,103
	Ni, Fe, Co	13 - 14	25 - 80	KOH Solution, AEM	
Hydrogen peroxide	Pt	0 - 1	25	H ₂ SO ₄ / HClO ₄ Solution, PEM	104,105
		13 - 14	25	KOH / NaOH Solution	
Acetaldehyde	Ti-Pt	0 - 1	25 - 80	H ₂ SO ₄ / HClO ₄ Solution	106,107
Acetic acid	Pd-TiO ₂ , FeOOH	0 - 1	25 - 80	H ₂ SO ₄ / HClO ₄ Solution	106-108
Ethyl acetate	Co ₃ O ₄ , PdCu	13 - 14	25	KOH Solution	109
Acrylic acid	Pt, Au, Pd-CeO ₂	0 - 1	25	HClO ₄ Solution	110
Lactic acid	Au, AuPt, Co	13 - 14	25	KOH Solution, AEM	111
Benzaldehyde	Cu-Co-N	-	45	Ionic Liquid [Bmin][BF ₄]	112,113
Benzoic acid	Ni	0 - 1	25 - 90	H ₂ SO ₄ Solution	114,115
2-Furoic acid	Ni ₃ S ₂	9.4	25 - 80	NaHCO ₃ /Na ₂ CO ₃	116,117
	Ni ₂ P/Ni	13 - 14	30	KOH Solution	

2,5-Furandicarboxylic acid (FDCA)	MnO ₂	1	60	H ₂ SO ₄ Solution	118,119
	NiFe	13 - 14	25	KOH Solution	120
4-Methoxybenzaldehyde	Graphite	-	25	TEMPO Solution	121
Acetophenone	-	-	25	CH ₂ Cl ₂ Solution	122,123
Acetone	NiS ₂	13 - 14	25	KOH Solution	124
Phenoxyacetic acid	Ni	13 - 14	25	NaOH Solution	125,126
Formaldehyde	-	0 - 1	25	HClO ₄ Solution	127,128
Formic acid	Pt	0 - 1	25	H ₂ SO ₄ Solution	129
Glycolic acid	Pd-TiO ₂	13 - 14	25	KOH Solution	129
Oxalic acid	FeCoNi	13 - 14	25	KOH Solution	129

* Mt: Mega Tonne, * PEM: Proton Exchange Membrane, * AEM: Anion Exchange Membrane

Figure 1b. I-V curves and required potentials at the cathode and anode for electrolysis.

Supplementary Table 3 Essential information of the oxidation products for the anode

Product	Half-cell reaction	E ⁰ (V) [*]	η (V) ^{**}	Market price (\$/kg)
Oxygen	2H ₂ O → O ₂ +4H ⁺ +4e ⁻	1.23	0.37 ^{148,***} 0.25 ¹⁴⁹	0.024-0.04 ¹⁵⁰
Hydrogen peroxide	2H ₂ O → H ₂ O ₂ +2H ⁺ +2e ⁻	1.78	0.72 ^{151,***} 0.42 ¹⁵²	0.56-0.58 ¹⁵³
Acetaldehyde	Ethanol → Acetaldehyde+2H ⁺ +2e ⁻	0.193	1.26 ¹⁵⁴	1.00 ¹⁵⁵
Acetic acid	Ethanol+H ₂ O → Acetic acid+4H ⁺ +4e ⁻	-0.334	1.78 ¹⁵⁶	0.68-0.92 ¹⁴³
Ethyl acetate	2Ethanol → Ethyl acetate+4H ⁺ +4e ⁻	-0.208	1.53 ¹⁵⁷	1.21-1.8 ¹⁵⁸
Acrylic acid	1,3-Propanediol → Acrylic acid+4H ⁺ +4e ⁻	0.248	0.55 ¹⁵⁹	2.25-2.88 ¹⁶⁰
Lactic acid	1,2-Propanediol+H ₂ O → Lactic acid+4H ⁺ +4e ⁻	-0.334	0.82 ¹¹¹	1.58-1.87 ¹⁵⁵

	Glycerol → Lactic acid+2H ⁺ +2e ⁻	0.041	0.41 ¹⁶¹	
Benzaldehyde	Benzyl alcohol → Benzaldehyde+2H ⁺ +2e ⁻	0.193	1.01 ¹⁶²	1.18-2.11 ¹⁶³
benzoic acid	Benzyl alcohol+H ₂ O → Benzoic acid+4H ⁺ +4e ⁻	-0.334	1.68 ³³	1.85 ¹⁶⁴
	Furfural+H ₂ O → 2-Furoic acid+2H ⁺ +2e ⁻	-1.27	2.7 ¹⁶⁵	
2-Furoic acid	Furfuryl alcohol+H ₂ O → 2-Furoic acid+4H ⁺ +4e ⁻	-0.515	1.88 ¹⁶⁷	6.23 ¹⁶⁶
2,5-Furandicarboxylic acid (FDCA)	5-Hydromethylfurfural (HMF)+H ₂ O → FDCA+6H ⁺ +6e ⁻	-0.78	2.03 ¹¹⁹	32 – 580 ¹⁶⁸
Acetone	Isopropanol+H ₂ O → Acetone+2H ⁺ +2e ⁻	0.054	1.30 ¹⁶⁹	0.9-1.28 ¹⁴²
Formaldehyde	Methanol → Formaldehyde+2H ⁺ +2e ⁻	0.465	0.04 ¹⁷⁰	0.37 - 0.74 ¹⁷¹
formic acid	Methanol+H ₂ O → Formic acid+4H ⁺ +4e ⁻	-0.258	0.76 ¹⁷²	0.97-1.08 ¹³²
Glycolic acid	Ethylene glycol+H ₂ O → Glycolic acid+4H ⁺ +4e ⁻	-0.334	0.67 ¹⁷³	1.84 ¹⁷⁴
Oxalic acid	Ethylene glycol+2H ₂ O → Oxalic acid+8H ⁺ +8e ⁻	-0.455	0.80 ¹⁷⁵	1.4 ¹⁷⁶

*Joback method¹⁷⁷ was used for unknown free energy, **Based on onset potential, ***used in this study

We also modified statement regarding the oxidation potential of OER according to Supplementary Table 3.

Changes made:

- In page 3 of supplementary information

Original sentence: “However, the OER is not an attractive reaction in terms of energy because oxygen is produced at high oxidative potential (normally > 1.45 V vs. RHE).”

Revision: “However, the OER is not an attractive reaction in terms of energy because oxygen is produced at high oxidative potential (normally > 1.48 vs. RHE, Supplementary Table 3).”

Reference

1. Liu W-J, Dang L, Xu Z, Yu H-Q, Jin S, Huber GW. Electrochemical oxidation of 5-Hydroxymethylfurfural with NiFe layered double hydroxide (LDH) nanosheet catalysts. *ACS Catalysis* **8**, 5533-5541 (2018).
2. Chen YX, Miki A, Ye S, Sakai H, Osawa M. Formate, an active intermediate for direct oxidation of methanol on Pt electrode. *Journal of the American Chemical Society* **125**, 3680-3681 (2003).

REVIEWERS' COMMENTS:

Reviewer #2 (Remarks to the Author):

The authors have dealt appropriately with my last concern. Both manuscript and supporting information are now suitable for publication.